# CRITICAL POINTS OF LINEAR NEURAL NETWORKS: ANALYTICAL FORMS AND LANDSCAPE PROPERTIES

**Yi Zhou & Yingbin Liang**
Department of Electrical and Computer Engineering
The Ohio State University
`zhou.1172@osu.edu`

## ABSTRACT

Due to the success of deep learning to solving a variety of challenging machine learning tasks, there is a rising interest in understanding loss functions for training neural networks from a theoretical aspect. Particularly, the properties of critical points and the landscape around them are of importance to determine the convergence performance of optimization algorithms. In this paper, we provide a necessary and sufficient characterization of the analytical forms for the critical points (as well as global minimizers) of the square loss functions for linear neural networks. We show that the analytical forms of the critical points characterize the values of the corresponding loss functions as well as the necessary and sufficient conditions to achieve global minimum. Furthermore, we exploit the analytical forms of the critical points to characterize the landscape properties for the loss functions of linear neural networks and shallow ReLU networks. One particular conclusion is that: While the loss function of linear networks has no spurious local minimum, the loss function of one-hidden-layer nonlinear networks with ReLU activation function does have local minimum that is not global minimum.

## 1  INTRODUCTION

In the past decade, deep neural networks Goodfellow et al. (2016) have become a popular tool that has successfully solved many challenging tasks in a variety of areas such as machine learning, artificial intelligence, computer vision, and natural language processing, etc. As the understandings of deep neural networks from different aspects are mostly based on empirical studies, there is a rising need and interest to develop understandings of neural networks from theoretical aspects such as generalization error, representation power, and landscape (also referred to as geometry) properties, etc. In particular, the landscape properties of loss functions (that are typically nonconex for neural networks) play a central role to determine the iteration path and convergence performance of optimization algorithms.

One major landscape property is the nature of critical points, which can possibly be global minima, local minima, saddle points. There have been intensive efforts in the past into understanding such an issue for various neural networks. For example, it has been shown that every local minimum of the loss function is also a global minimum for shallow linear networks under the autoencoder setting and invertibility assumptions Baldi & Hornik (1989) and for deep linear networks Kawaguchi (2016); Lu & Kawaguchi (2017); Yun et al. (2017) respectively under different assumptions. The conditions on the equivalence between local minimum or critical point and global minimum has also been established for various nonlinear neural networks Yu & Chen (1995); Gori & Tesi (1992); Nguyen & Hein (2017); Soudry & Carmon (2016); Feizi et al. (2017) under respective assumptions.

However, most previous studies did not provide characterization of analytical forms for critical points of loss functions for neural networks with only very few exceptions. In Baldi & Hornik (1989), the authors provided an analytical form for the critical points of the square loss function of shallow linear networks under certain conditions. Such an analytical form further helps to establish the landscape properties around the critical points. Further in Li et al. (2016b), the authors characterized certain sufficient form of critical points for the square loss function of matrix factorization problems and deep linear networks.

The focus of this paper is on characterizing the sufficient and necessary forms of critical points for broader scenarios, i.e., shallow and deep linear networks with no assumptions on data matrices and network dimensions, and shallow ReLU networks over certain parameter space. In particular, such analytical forms of critical points capture the corresponding loss function values and the necessary and sufficient conditions to achieve global minimum. This further enables us to establish new landscape properties around these critical points for the loss function of these networks under general settings, and provides alternative (yet simpler and more intuitive) proofs for existing understanding of the landscape properties.

OUR CONTRIBUTION

1) For the square loss function of linear networks with one hidden layer, we provide a full (necessary and sufficient) characterization of the analytical forms for its critical points and global minimizers. These results generalize the characterization in Baldi & Hornik (1989) to arbitrary network parameter dimensions and any data matrices. Such a generalization further enables us to establish the landscape property, i.e., every local minimum is also a global minimum and all other critical points are saddle points, under no assumptions on parameter dimensions and data matrices. From a technical standpoint, we exploit the analytical forms of critical points to provide a new proof for characterizing the landscape around the critical points under full relaxation of assumptions, where the corresponding approaches in Baldi & Hornik (1989) are not applicable. As a special case of linear networks, the matrix factorization problem satisfies all these landscape properties.

2) For the square loss function of deep linear networks, we establish a full (necessary and sufficient) characterization of the analytical forms for its critical points and global minimizers. Such characterizations are new and have not been established in the existing art. Furthermore, such analytical form divides the set of non-global-minimum critical points into different categories. We identify the directions along which the loss function value decreases for two categories of the critical points, for which our result directly implies the equivalence between the local minimum and the global minimum. For these cases, our proof generalizes the result in Kawaguchi (2016) under no assumptions on the network parameter dimensions and data matrices.

3) For the square loss function of one-hidden-layer nonlinear neural networks with ReLU activation function, we provide a full characterization of both the existence and the analytical forms of the critical points in certain types of regions in the parameter space. Particularly, in the case where there is one hidden unit, our results fully characterize the existence and the analytical forms of the critical points in the entire parameter space. Such characterization were not provided in previous work on nonlinear neural networks. Moreover, we apply our results to a concrete example to demonstrate that both local minimum that is not a global minimum and local maximum do exist in such a case.

RELATED WORK

**Analytical forms of critical points**: Characterizing the analytical form of critical points for loss functions of neural networks dates back to Baldi & Hornik (1989), where the authors provided an analytical form of the critical points for the square loss function of linear networks with one hidden layer. In Li et al. (2016b), the authors provided a sufficient condition of critical points of a generic function, i.e., the fixed point of invariant groups. They then characterized certain sufficient forms of critical points for the square loss function of matrix factorization problems and deep linear networks, whereas our results provide sufficient and necessary forms of critical points for deep linear networks via a different approach.

**Properties of critical points**: Baldi & Hornik (1989); Baldi (1989) studied the linear autoencoder with one hidden layer and showed the equivalence between the local minimum and the global minimum. Moreover, Baldi & Lu (2012) generalized these results to the complex-valued autoencoder setting. The deep linear networks were studied by some recent work Kawaguchi (2016); Lu & Kawaguchi (2017); Yun et al. (2018), in which the equivalence between the local minimum and the global minimum was established respectively under different assumptions. Particularly, Yun et al. (2017) established a necessary and sufficient condition for a critical point of the deep linear network to be a global minimum. A similar result was established in Freeman & Bruna (2017) for deep linear networks under the setting that the widths of intermediate layers are larger than those of the input

and output layers. The effect of regularization on the critical points for a two-layer linear network was studied in Taghvaei et al. (2017).

For nonlinear neural networks, Yu & Chen (1995) studied a nonlinear neural network with one hidden layer and sigmoid activation function, and showed that every local minimum is also a global minimum provided that the number of input units equals the number of data samples. Gori & Tesi (1992) considered a class of multi-layer nonlinear networks with a pyramidal structure, and showed that all critical points of full column rank achieve the zero loss when the sample size is less than the input dimension. These results were further generalized to a larger class of nonlinear networks in Nguyen & Hein (2017), in which they also showed that critical points with non-degenerate Hessian are global minimum. Choromanska et al. (2015a;b) connected the loss surface of deep nonlinear networks with the Hamiltonian of the spin-glass model under certain assumptions and characterized the distribution of the local minimum. Kawaguchi (2016) further eliminated some of the assumptions in Choromanska et al. (2015a), and established the equivalence between the local minimum and the global minimum by reducing the loss function of the deep nonlinear network to that of the deep linear network. Soudry & Carmon (2016) showed that a two-layer nonlinear network has no bad differentiable local minimum. Feizi et al. (2017) studied a one-hidden-layer nonlinear neural network with the parameters restricted in a set of directions of lines, and showed that most local minima are global minima. Tian (2017) considered a two-layer ReLU network with Gaussian input data, and showed that critical points in certain region are non-isolated and characterized the critical-point-free regions.

**Geometric curvature**: Hardt & Ma (2017) established the gradient dominance condition of deep linear residual networks, and Zhou & Liang (2017) further established the gradient dominance condition and regularity condition around the global minimizers for deep linear, deep linear residual and shallow nonlinear networks. Li et al. (2016a) studied the property of the Hessian matrix for deep linear residual networks. The local strong convexity property was established in Soltanolkotabi et al. (2017) for overparameterized nonlinear networks with one hidden layer and quadratic activation functions, and was established in Zhong et al. (2017) for a class of nonlinear networks with one hidden layer and Gaussian input data. Zhong et al. (2017) further established the local linear convergence of gradient descent method with tensor initialization. Soudry & Hoffer (2017) studied a one-hidden-layer nonlinear network with a single output, and showed that the volume of sub-optimal differentiable local minima is exponentially vanishing in comparison with the volume of global minima. Dauphin et al. (2014) investigated the saddle points in deep neural networks using the results from statistical physics and random matrix theory.

**Notation:** The pseudoinverse, column space and null space of a matrix $\boldsymbol{M}$ are denoted by $\boldsymbol{M}^\dagger, \mathrm{col}(\boldsymbol{M})$ and $\ker(\boldsymbol{M})$, respectively. For any index sets $I, J \subset \mathbb{N}$, $\boldsymbol{M}_{I,J}$ denotes the submatrix of $\boldsymbol{M}$ formed by the entries with the row indices in $I$ and the column indices in $J$. For positive integers $i \leq j$, we define $i : j = \{i, i+1, \ldots, j-1, j\}$. The projection operator onto a linear subspace $V$ is denoted by $\mathcal{P}_V$.

## 2 LINEAR NEURAL NETWORKS WITH ONE HIDDEN LAYER

In this section, we study linear neural networks with one hidden layer. Suppose we have an input data matrix $\boldsymbol{X} \in \mathbb{R}^{d_0 \times m}$ and a corresponding output data matrix $\boldsymbol{Y} \in \mathbb{R}^{d_2 \times m}$, where there are in total $m$ data samples. We are interested in learning a model that maps from $\boldsymbol{X}$ to $\boldsymbol{Y}$ via a linear network with one hidden layer. Specifically, we denote the weight parameters between the output layer and the hidden layer of the network as $\boldsymbol{A}_2 \in \mathbb{R}^{d_2 \times d_1}$, and denote the weight parameters between the hidden layer and the input layer of the network as $\boldsymbol{A}_1 \in \mathbb{R}^{d_1 \times d_0}$. We are interested in the square loss function of this linear network, which is given by

$$\mathcal{L} := \tfrac{1}{2} \left\| \boldsymbol{A}_2 \boldsymbol{A}_1 \boldsymbol{X} - \boldsymbol{Y} \right\|_F^2.$$

Note that in a special case where $\boldsymbol{X} = \boldsymbol{I}$, $\mathcal{L}$ reduces to a loss function for the matrix factorization problem, to which all our results apply. The loss function $\mathcal{L}$ has been studied in Baldi & Hornik (1989) under the assumptions that $d_2 = d_0 \geq d_1$ and the matrices $\boldsymbol{X}\boldsymbol{X}^\top, \boldsymbol{Y}\boldsymbol{X}^\top(\boldsymbol{X}\boldsymbol{X}^\top)^{-1}\boldsymbol{X}\boldsymbol{Y}^\top$ are invertible. In our study, no assumption is made on either the parameter dimensions or the invertibility of the data matrices. Such full generalization of the results in Baldi & Hornik (1989) turns out to be critical for our study of nonlinear shallow neural networks in Section 4.

We further define $\boldsymbol{\Sigma} := \boldsymbol{YX}^\dagger \boldsymbol{XY}^\top$ and denote its full singular value decomposition as $\boldsymbol{U\Lambda U}^\top$. Suppose that $\boldsymbol{\Sigma}$ has $r$ distinct positive singular values $\sigma_1 > \cdots > \sigma_r > 0$ with multiplicities $m_1, \ldots, m_r$, respectively, and has $\bar{m}$ zero singular values. Recall that $(\boldsymbol{A}_1, \boldsymbol{A}_2)$ is defined to be a critical point of $\mathcal{L}$ if $\nabla_{\boldsymbol{A}_1}\mathcal{L} = \boldsymbol{0}, \nabla_{\boldsymbol{A}_2}\mathcal{L} = \boldsymbol{0}$. Our first result provides a full characterization of all critical points of $\mathcal{L}$.

**Theorem 1** (Characterization of critical points). *All critical points of $\mathcal{L}$ are necessarily and sufficiently characterized by a matrix $\boldsymbol{L}_1 \in \mathbb{R}^{d_1 \times d_0}$, a block matrix $\boldsymbol{V} \in \mathbb{R}^{d_2 \times d_1}$ and an invertible matrix $\boldsymbol{C} \in \mathbb{R}^{d_1 \times d_1}$ via*

$$\boldsymbol{A}_1 = \boldsymbol{C}^{-1}\boldsymbol{V}^\top \boldsymbol{U}^\top \boldsymbol{YX}^\dagger + \boldsymbol{L}_1 - \boldsymbol{C}^{-1}\boldsymbol{V}^\top \boldsymbol{VC}\boldsymbol{L}_1 \boldsymbol{XX}^\dagger \tag{1}$$

$$\boldsymbol{A}_2 = \boldsymbol{UVC}. \tag{2}$$

*Specifically, $\boldsymbol{V} = [\mathrm{diag}(\boldsymbol{V}_1, \ldots, \boldsymbol{V}_r, \overline{\boldsymbol{V}}), \boldsymbol{0}^{d_2 \times (d_1 - \mathrm{rank}(\boldsymbol{A}_2))}]$, where both $\boldsymbol{V}_i \in \mathbb{R}^{m_i \times p_i}$ and $\overline{\boldsymbol{V}} \in \mathbb{R}^{\bar{m} \times \bar{p}}$ consist of orthonormal columns with the number of columns $p_i \leq m_i, i = 1, \ldots, r, \bar{p} \leq \bar{m}$ such that $\sum_{i=1}^r p_i + \bar{p} = \mathrm{rank}(\boldsymbol{A}_2)$, and $\boldsymbol{L}_1, \boldsymbol{V}, \boldsymbol{C}$ satisfy*

$$\mathcal{P}_{\mathrm{col}(\boldsymbol{UV})^\perp} \boldsymbol{YX}^\top \boldsymbol{L}_1^\top \boldsymbol{C}^\top \mathcal{P}_{\mathrm{ker}(\boldsymbol{V})} = \boldsymbol{0}. \tag{3}$$

Theorem 1 characterizes the necessary and sufficient forms for all critical points of $\mathcal{L}$. Intuitively, the matrix $\boldsymbol{C}$ captures the invariance of the product $\boldsymbol{A}_2\boldsymbol{A}_1$ under an invertible transform, and $\boldsymbol{L}_1$ captures the degree of freedom of the solution set for linear systems.

In general, the set of critical points is uncountable and cannot be fully listed out. However, the analytical forms in eqs. (1) and (2) do allow one to construct some critical points of $\mathcal{L}$ by specifying choices of $\boldsymbol{L}_1, \boldsymbol{V}, \boldsymbol{C}$ that fulfill the condition in eq. (3). For example, choosing $\boldsymbol{L}_1 = \boldsymbol{0}$ guarantees eq. (3), in which case eqs. (1) and (2) yield a critical point $(\boldsymbol{C}^{-1}\boldsymbol{V}^\top\boldsymbol{U}^\top\boldsymbol{YX}^\dagger, \boldsymbol{UVC})$ for any invertible matrix $\boldsymbol{C}$ and any block matrix $\boldsymbol{V}$ that takes the form specified in Theorem 1. For nonzero $\boldsymbol{L}_1$, one can fix a proper $\boldsymbol{V}$ and solve the linear equation on $\boldsymbol{C}$ in eq. (3). If a solution exists, we then obtain the form of a corresponding critical point. We further note that the analytical structures of the critical points are more important, which have direct implications on the global optimality conditions and landscape properties as we show in the remaining part of the section.

**Remark 1.** *We note that the block pattern parameters $\{p_i\}_{i=1}^r$ and $\bar{p}$ denote the number of columns of $\{\boldsymbol{V}_i\}_{i=1}^r$ and $\overline{\boldsymbol{V}}$, respectively, and their sum equals the rank of $\boldsymbol{A}_2$, i.e., $\sum_{i=1}^r p_i + \bar{p} = \mathrm{rank}(\boldsymbol{A}_2)$.*

The parameters $p_i, i = 1, \ldots, r, \bar{p}$ of $\boldsymbol{V}$ contain all useful information of the critical points that determine the function value of $\mathcal{L}$ as presented in the following proposition.

**Proposition 1.** *Any critical point $(\boldsymbol{A}_1, \boldsymbol{A}_2)$ of $\mathcal{L}$ satisfies $\mathcal{L}(\boldsymbol{A}_1, \boldsymbol{A}_2) = \frac{1}{2}(\mathrm{Tr}(\boldsymbol{YY}^\top) - \sum_{i=1}^r p_i\sigma_i)$.*

Proposition 1 evaluates the function value $\mathcal{L}$ at a critical point using the parameters $\{p_i\}_{i=1}^r$. To explain further, recall that the data matrix $\boldsymbol{\Sigma}$ has each singular value $\sigma_i$ with multiplicity $m_i$. For each $i$, the critical point captures $p_i$ out of $m_i$ singular values $\sigma_i$. Hence, for a $\sigma_i$ with larger value (i.e., a smaller index $i$), it is desirable that a critical point captures a larger number $p_i$ of them. In this way, the critical point captures more important principle components of the data so that the value of the loss function is further reduced as suggested by Proposition 1. In summary, the parameters $\{p_i\}_{i=1}^r$ characterize how well the learned model fits the data in terms of the value of the loss function. Moreover, the parameters $\{p_i\}_{i=1}^r$ also determine a full characterization of the global minimizers as given below.

**Proposition 2** (Characterization of global minimizers). *A critical point $(\boldsymbol{A}_1, \boldsymbol{A}_2)$ of $\mathcal{L}$ is a global minimizer if and only if it falls into the following two cases.*

1. *Case 1: $\min\{d_2, d_1\} \leq \sum_{i=1}^r m_i$, $\boldsymbol{A}_2$ is full rank, and $p_1 = m_1, \ldots, p_{k-1} = m_{k-1}, p_k = \mathrm{rank}(\boldsymbol{A}_2) - \sum_{i=1}^{k-1} m_i \leq m_k$ for some $k \leq r$;*
2. *Case 2: $\min\{d_2, d_1\} > \sum_{i=1}^r m_i$, $p_i = m_i$ for $i = 1, \ldots, r$, and $\bar{p} \geq 0$. In particular, $\boldsymbol{A}_2$ can be non-full rank with $\mathrm{rank}(\boldsymbol{A}_2) = \sum_{i=1}^r m_i$.*

*The analytical form of any global minimizer can be obtained from Theorem 1 with further specification to the above two cases.*

Proposition 2 establishes the neccessary and sufficient conditions for any critical point to be a global minimizer. If the data matrix $\boldsymbol{\Sigma}$ has a large number of nonzero singular values, i.e., the first case,

one needs to exhaust the representation budget (i.e., rank) of $A_2$ and capture as many large singular values as the rank allows to achieve the global minimum; Otherwise, $A_2$ of a global minimizer can be non-full rank and still captures all nonzero singular values. Note that $A_2$ must be full rank in the case 1, and so is $A_1$ if we further adopt the assumptions on the network size and data matrices in Baldi & Hornik (1989).

Furthermore, the parameters $\{p_i\}_{i=1}^r$ naturally divide all non-global-minimum critical points $(A_1, A_2)$ of $\mathcal{L}$ into the following two categories.

- (Non-optimal order): The matrix $V$ specified in Theorem 1 satisfies that there exists $1 \leq i < j \leq r$ such that $p_i < m_i$ and $p_j > 0$.
- (Optimal order): $\text{rank}(A_2) < \min\{d_2, d_1\}$ and the matrix $V$ specified in Theorem 1 satisfies that $p_1 = m_1, \ldots, p_{k-1} = m_{k-1}, p_k = \text{rank}(A_2) - \sum_{i=1}^{k-1} m_i \leq m_k$ for some $1 \leq k \leq r$.

To understand the above two categories, note that a critical point of $\mathcal{L}$ with non-optimal order captures a smaller singular value $\sigma_j$ (since $p_j > 0$) while skipping a larger singular value $\sigma_i$ with a lower index $i < j$ (since $p_i < m_i$), and hence cannot be a global minimizer. On the other hand, although a critical point of $\mathcal{L}$ with optimal order captures the singular values in the optimal (i.e., decreasing) order, it does not fully utilize the representation budget of $A_2$ (because $A_2$ is non-full rank) to further capture nonzero singular values and reduce the function value, and hence cannot be a global minimizer either. Next, we show that these two types of non-global-minimum critical points have different landscape properties around them. Throughout, a matrix $\widetilde{M}$ is called the perturbation of $M$ if it lies in an arbitrarily small neighborhood of $M$.

**Proposition 3** (Landscape around critical points). *The critical points of $\mathcal{L}$ have the following landscape properties.*

1. *A non-optimal-order critical point $(A_1, A_2)$ has a perturbation $(\widetilde{A_1}, \widetilde{A_2})$ with $\text{rank}(\widetilde{A_2}) = \text{rank}(A_2)$, which achieves a lower function value;*
2. *An optimal-order critical point $(A_1, A_2)$ has a perturbation $(\widetilde{A_1}, \widetilde{A_2})$ with $\text{rank}(\widetilde{A_2}) = \text{rank}(A_2) + 1$, which achieves a lower function value;*
3. *Any point in $\mathcal{X} := \{(A_1, A_2) : A_2 A_1 X \neq 0\}$ has a perturbation $(A_1, \widetilde{A_2})$, which achieves a higher function value;*

As a consequence, items 1 and 2 imply that any non-global-minimum critical point has a descent direction, and hence cannot be a local minimizer. Thus, any local minimizer must be a global minimizer. Item 3 implies that any point has an ascent direction whenever the output is nonzero. Hence, there does not exist any local/global maximizer in $\mathcal{X}$. Furthermore, item 3 together with items 1 and 2 implies that any non-global-minimum critical point in $\mathcal{X}$ has both descent and ascent directions, and hence must be a saddle point. We summarize these facts in the following theorem.

**Theorem 2** (Landscape of $\mathcal{L}$). *The loss function $\mathcal{L}$ satisfies: 1) every local minimum is also a global minimum; 2) every non-global-minimum critical point in $\mathcal{X}$ is a saddle point.*

We note that the saddle points in Theorem 2 can be non-strict when the data matrices are singular. As an illustrative example, consider the following loss function of a shallow linear network $\mathcal{L}(a_2, a_1) = \frac{1}{2}(a_2 a_1 x - y)^2$, where $a_1, a_2, x$ and $y$ are all scalars. Consider the case $y = 0$. Then, the Hessian at the saddle point $a_1 = 0, a_2 = 1$ is $[x^2, 0; 0, 0]$, which does not have any negative eigenvalue.

From a technical point of view, the proof of item 1 of Proposition 3 applies that in Baldi (1989) and generalizes it to the setting where $\Sigma$ can have repeated singular values and may not be invertible. To further understand the perturbation scheme from a high level perspective, note that non-optimal-order critical points capture a smaller singular value $\sigma_j$ instead of a larger one $\sigma_i$ with $i < j$. Thus, one naturally perturbs the singular vector corresponding to $\sigma_j$ along the direction of the singular vector corresponding to $\sigma_i$. Such a perturbation scheme preserves the rank of $A_2$ and reduces the value of the loss function.

More importantly, the proof of item 2 of Proposition 3 introduces a new technique. As a comparison, Baldi & Hornik (1989) proves a similar result as item 2 using the strict convexity of the function, which requires the parameter dimensions to satisfy $d_2 = d_0 \geq d_1$ and the data matrices to be invertible. In contrast, our proof completely removes these restrictions by introducing a new perturbation direction and exploiting the analytical forms of critical points in eqs. (1) and (2) and the condition in

eq. (3). The accomplishment of the proof further requires careful choices of perturbation parameters as well as judicious manipulations of matrices. We refer the reader to the supplemental materials for more details. As a high level understanding, since optimal-order critical points capture the singular values in an optimal (i.e., decreasing) order, the previous perturbation scheme for non-optimal-order critical points does not apply. Instead, we increase the rank of $\boldsymbol{A}_2$ by one in a way that the perturbed matrix captures the next singular value beyond the ones that have already been captured so that the value of the loss function can be further reduced.

## 3 Deep Linear Neural Networks

In this section, we study deep linear networks with $\ell \geq 2$ layers. We denote the weight parameters between the layers as $\boldsymbol{A}_k \in \mathbb{R}^{d_k \times d_{k-1}}$ for $k = 1, \ldots, \ell$, respectively. The input and output data are denoted by $\boldsymbol{X} \in \mathbb{R}^{d_0 \times m}, \boldsymbol{Y} \in \mathbb{R}^{d_\ell \times m}$, respectively. We are interested in the square loss function of deep linear networks, which is given by

$$\mathcal{L}_D := \tfrac{1}{2} \left\| \boldsymbol{A}_\ell \cdots \boldsymbol{A}_2 \boldsymbol{A}_1 \boldsymbol{X} - \boldsymbol{Y} \right\|_F^2.$$

Denote $\boldsymbol{\Sigma}_k := \boldsymbol{Y}(\boldsymbol{A}_{(k,1)}\boldsymbol{X})^\dagger \boldsymbol{A}_{(k,1)} \boldsymbol{X} \boldsymbol{Y}^\top$ for $k = 0, \ldots, \ell$ with the full singular value decomposition $\boldsymbol{U}_k \boldsymbol{\Lambda}_k \boldsymbol{U}_k^\top$. Suppose that $\boldsymbol{\Sigma}_k$ has $r(k)$ distinct positive singular values $\sigma_1(k) > \cdots > \sigma_{r(k)}(k) > 0$ with multiplicities $m_1(k), \ldots, m_{r(k)}(k)$, respectively, and $\bar{m}(k)$ zero singular values. Our first result provides a full characterization of all critical points of $\mathcal{L}_D$, where we denote $\boldsymbol{A}_{(i,j)} := \boldsymbol{A}_i \boldsymbol{A}_{i-1} \cdots \boldsymbol{A}_{j+1} \boldsymbol{A}_j$ for notational convenience[1].

**Theorem 3** (Characterization of critical points). *All critical points of $\mathcal{L}_D$ are necessarily and sufficiently characterized by matrices $\boldsymbol{L}_k \in \mathbb{R}^{d_k \times d_{k-1}}$, block matrices $\boldsymbol{V}_k \in \mathbb{R}^{d_l \times d_{k+1}}$ and invertible matrices $\boldsymbol{C}_k \in \mathbb{R}^{d_{k+1} \times d_{k+1}}$ for $k = 0, \ldots, \ell-2$ such that $\boldsymbol{A}_1, \ldots, \boldsymbol{A}_\ell$ can be individually expressed out recursively via the following two equations:*

$$\boldsymbol{A}_{k+1} = \boldsymbol{C}_k^{-1} \boldsymbol{V}_k^\top \boldsymbol{U}_k^\top \boldsymbol{Y}(\boldsymbol{A}_{(k,1)}\boldsymbol{X})^\dagger + \boldsymbol{L}_{k+1} - \boldsymbol{C}_k^{-1}\boldsymbol{V}_k^\top \boldsymbol{V}_k \boldsymbol{C}_k \boldsymbol{L}_{k+1} \boldsymbol{A}_{(k,1)} \boldsymbol{X}(\boldsymbol{A}_{(k,1)}\boldsymbol{X})^\dagger, \quad (4)$$

$$\boldsymbol{A}_{(\ell,k+2)} = \boldsymbol{U}_k \boldsymbol{V}_k \boldsymbol{C}_k. \quad (5)$$

*Specifically,* $\boldsymbol{V}_k = [\mathrm{diag}(\boldsymbol{V}_1^{(k)}, \ldots, \boldsymbol{V}_{r(k)}^{(k)}, \overline{\boldsymbol{V}}^{(k)}), \boldsymbol{0}^{d_l \times (d_{k+1} - \mathrm{rank}(\boldsymbol{A}_{(\ell,k+2)}))}]$, *where* $\boldsymbol{V}_i^{(k)} \in \mathbb{R}^{m_i(k) \times p_i(k)}, \overline{\boldsymbol{V}}^{(k)} \in \mathbb{R}^{\bar{m}(k) \times \bar{p}(k)}$ *consist of orthonormal columns with* $p_i(k) \leq m_i(k)$ *for* $i = 1, \ldots, r(k)$, $\bar{p}(k) \leq \bar{m}(k)$ *such that* $\sum_{i=1}^{r(k)} p_i(k) + \bar{p}(k) = \mathrm{rank}(\boldsymbol{A}_{\ell,k+1})$, *and* $\boldsymbol{L}_k, \boldsymbol{V}_k, \boldsymbol{C}_k$ *satisfy for* $k = 2, \ldots, \ell-1$

$$\boldsymbol{A}_{(\ell,k)} = \boldsymbol{A}_{(\ell,k+1)}\boldsymbol{A}_k, \quad (\boldsymbol{I} - \mathcal{P}_{\mathrm{col}(\boldsymbol{A}_{(\ell,k+1)})})\boldsymbol{Y}\boldsymbol{X}^\top \boldsymbol{A}_{(\ell-1,1)}^\top = \boldsymbol{0}. \quad (6)$$

Note that the forms of the individual parameters $\boldsymbol{A}_1, \ldots, \boldsymbol{A}_\ell$ can be obtained as follows by recursively applying eqs. (4) and (5). First, eq. (5) with $k = 0$ yields the form of $\boldsymbol{A}_{(\ell,2)}$. Then, eq. (4) with $k = 0$ and the form of $\boldsymbol{A}_{(\ell,2)}$ yield the form of $\boldsymbol{A}_1$. Next, eq. (5) with $k = 1$ yields the form of $\boldsymbol{A}_{(\ell,3)}$, and then, eq. (4) with $k = 1$ and the forms of $\boldsymbol{A}_{(\ell,3)}, \boldsymbol{A}_1$ further yield the form of $\boldsymbol{A}_2$. Inductively, one obtains the expressions of all individual parameter matrices. Furthermore, the first condition in eq. (6) is a consistency condition that guarantees that the analytical form for the entire product of parameter matrices factorizes into the forms of individual parameter matrices. Similarly to shallow linear networks, while the set of critical points here is also uncountable, Theorem 3 suggests ways to obtain some critical points. For example, if we set $\boldsymbol{L}_k = \boldsymbol{0}$ for all $k$ (i.e., eq. (6) is satisfied), we can obtain the form of critical points for any invertible $\boldsymbol{C}_k$ and proper $\boldsymbol{V}_k$ with the structure specified in Theorem 3. For nonzero $\boldsymbol{L}_k$, eq. (6) needs to be verified for given $\boldsymbol{C}_k$ and $\boldsymbol{V}_k$ to determine a critical point.

Similarly to shallow linear networks, the parameters $\{p_i(0)\}_{i=1}^{r(0)}, \bar{p}(0)$ determine the value of the loss function at the critical points and further specify the analytical form for the global minimizers, as we present in the following two propositions.

**Proposition 4.** *Any critical point $(\boldsymbol{A}_1, \ldots, \boldsymbol{A}_\ell)$ of $\mathcal{L}_D$ satisfies*

$$\mathcal{L}_D(\boldsymbol{A}_1, \ldots, \boldsymbol{A}_\ell) = \tfrac{1}{2}\big[\mathrm{Tr}(\boldsymbol{Y}\boldsymbol{Y}^\top) - \sum_{i=1}^{r(0)} p_i(0)\sigma_i(0)\big].$$

---

[1]Here, $\boldsymbol{A}_{(0,1)}$ should be understood as identity matrix $\boldsymbol{I}$.

**Proposition 5** (Characterization of global minimizers). *A critical point* $(\boldsymbol{A}_1, \ldots, \boldsymbol{A}_\ell)$ *of* $\mathcal{L}_D$ *is a global minimizer if and only if it falls into the following two cases.*

1. *Case 1:* $\min\{d_\ell, \ldots, d_1\} \leq \sum_{i=1}^{r(0)} m_i(0)$, $\boldsymbol{A}_{(\ell,2)}$ *achieves the maximal rank, and* $p_1(0) = m_1(0), \ldots, p_{k-1}(0) = m_{k-1}(0), p_k(0) = \mathrm{rank}(\boldsymbol{A}_{(\ell,2)}) - \sum_{i=1}^{k-1} m_i(0) \leq m_k(0)$ *for some* $k \leq r(0)$;

2. *Case 2:* $\min\{d_\ell, \ldots, d_1\} > \sum_{i=1}^{r(0)} m_i(0)$, $p_i(0) = m_i(0)$ *for all* $i = 1, \ldots, r(0)$ *and* $\bar{p}(0) \geq 0$. *In particular,* $\boldsymbol{A}_{(\ell,2)}$ *can be non-full rank with* $\mathrm{rank}(\boldsymbol{A}_{(\ell,2)}) = \sum_{i=1}^{r(0)} m_i(0)$.

*The analytical form of any global minimizer can be obtained from Theorem 3 with further specification to the above two cases.*

In particular for case 1, if we further adopt the invertibility assumptions on data matrices as in Baldi & Hornik (1989) and assume that all parameter matrices are square, then all global minima must correspond to full rank parameter matrices.

We next exploit the analytical forms of the critical points to further understand the landscape of the loss function $\mathcal{L}_D$. It has been shown in Kawaguchi (2016) that every local minimum of $\mathcal{L}_D$ is also a global minimum, under certain conditions on the parameter dimensions and the invertibility of the data matrices. Here, our characterization of the analytical forms for the critical points allow us to understand such a result from an alternative viewpoint. The proofs for certain cases (that we discuss below) are simpler and more intuitive, and no assumption is made on the data matrices and dimensions of the network.

Similarly to shallow linear networks, we want to understand the local landscape around the critical points. However, due to the effect of depth, the critical points of $\mathcal{L}_D$ are more complicated than those of $\mathcal{L}$. Among them, we identify the following subsets of the non-global-minimum critical points $(\boldsymbol{A}_1, \cdots, \boldsymbol{A}_\ell)$ of $\mathcal{L}_D$.

- (Deep-non-optimal order): There exist $0 \leq k \leq \ell - 2$ such that the matrix $\boldsymbol{V}_k$ specified in Theorem 3 satisfies that there exist $1 \leq i < j \leq r(k)$ such that $p_i(k) < m_i(k)$ and $p_j(k) > 0$.
- (Deep-optimal order): $(\boldsymbol{A}_\ell, \boldsymbol{A}_{\ell-1})$ is not a global minimizer of $\mathcal{L}_D$ with $\boldsymbol{A}_{(\ell-2,1)}$ being fixed, $\mathrm{rank}(\boldsymbol{A}_\ell) < \min\{d_\ell, d_{\ell-1}\}$, and the matrix $\boldsymbol{V}_{\ell-2}$ specified in Theorem 3 satisfies that $p_1(l-2) = m_1(l-2), \ldots, p_{k-1}(l-2) = m_{k-1}(l-2), p_k(l-2) = \mathrm{rank}(\boldsymbol{A}_l) - \sum_{i=1}^{k-1} m_i(l-2) \leq m_k(l-2)$ for some $1 \leq k \leq r(l-2)$.

The following result summarizes the landscape of $\mathcal{L}_D$ around the above two types of critical points.

**Theorem 4** (Landscape of $\mathcal{L}_D$). *The loss function* $\mathcal{L}_D$ *has the following landscape properties.*

1. *A deep-non-optimal-order critical point* $(\boldsymbol{A}_1, \ldots, \boldsymbol{A}_\ell)$ *has a perturbation* $(\boldsymbol{A}_1, \ldots, \widetilde{\boldsymbol{A}}_{k+1}, \ldots, \widetilde{\boldsymbol{A}}_\ell)$ *with* $\mathrm{rank}(\widetilde{\boldsymbol{A}}_\ell) = \mathrm{rank}(\boldsymbol{A}_\ell)$, *which achieves a lower function value.*

2. *A deep-optimal-order critical point* $(\boldsymbol{A}_1, \ldots, \boldsymbol{A}_\ell)$ *has a perturbation* $(\boldsymbol{A}_1, \ldots, \widetilde{\boldsymbol{A}}_{\ell-1}, \widetilde{\boldsymbol{A}}_\ell)$ *with* $\mathrm{rank}(\widetilde{\boldsymbol{A}}_\ell) = \mathrm{rank}(\boldsymbol{A}_\ell) + 1$, *which achieves a lower function value.*

3. *Any point in* $\mathcal{X}_D := \{(\boldsymbol{A}_1, \ldots, \boldsymbol{A}_\ell) : \boldsymbol{A}_{(\ell,1)} \boldsymbol{X} \neq \boldsymbol{0}\}$ *has a perturbation* $(\boldsymbol{A}_1, \ldots, \widetilde{\boldsymbol{A}}_\ell)$ *that achieves a higher function value.*

*Consequently, 1) every local minimum of* $\mathcal{L}_D$ *is also a global minimum for the above two types of critical points; and 2) every critical point of these two types in* $\mathcal{X}_D$ *is a saddle point.*

Theorem 4 implies that the landscape of $\mathcal{L}_D$ for deep linear networks is similar to that of $\mathcal{L}$ for shallow linear networks, i.e., the pattern of the parameters $\{p_i(k)\}_{i=1}^{r(k)}$ implies different descent directions of the function value around the critical points. Our approach does not handle the remaining set of non-global minimizers, i.e., there exists $q \leq \ell-1$ such that $(\boldsymbol{A}_\ell, \ldots, \boldsymbol{A}_q)$ is a global minimum point of $\mathcal{L}_D$ with $\boldsymbol{A}_{(q-1,1)}$ being fixed, and $\boldsymbol{A}_{(\ell,q)}$ is of optimal order. It is unclear how to perturb the intermediate weight parameters using their analytical forms for deep networks , and we leave this as an open problem for the future work.

# 4 Nonlinear Neural Networks with One Hidden Layer

In this section, we study nonlinear neural networks with one hidden layer. In particular, we consider nonlinear networks with ReLU activation function $\sigma : \mathbb{R} \to \mathbb{R}$ that is defined as $\sigma(x) := \max\{x, 0\}$. Our study focuses on the set of differentiable critical points. The weight parameters between the layers are denoted by $\boldsymbol{A}_2 \in \mathbb{R}^{d_2 \times d_1}, \boldsymbol{A}_1 \in \mathbb{R}^{d_1 \times d_0}$, respectively, and the input and output data are denoted by $\boldsymbol{X} \in \mathbb{R}^{d_0 \times m}, \boldsymbol{Y} \in \mathbb{R}^{d_2 \times m}$, respectively. We are interested in the square loss function which is given by

$$\mathcal{L}_N := \tfrac{1}{2} \left\| \boldsymbol{A}_2 \sigma(\boldsymbol{A}_1 \boldsymbol{X}) - \boldsymbol{Y} \right\|_F^2, \tag{7}$$

where $\sigma$ acts on $\boldsymbol{A}_1 \boldsymbol{X}$ entrywise. Existing studies on nonlinear networks characterized the sufficient conditions for critical points being global minimum Gori & Tesi (1992); Nguyen & Hein (2017), established the equivalence between local minimum and global minimum under the condition that $d_0 = m$ Yu & Chen (1995), and provided understanding of geometric properties of the critical points Tian (2017). In comparison, our results below provide a full characterization of the critical points of $\mathcal{L}_N$ with $d_1 = 1$ and critical points of $\mathcal{L}_N$ over certain parameter space with $d_1 > 1$, and show that local minimum of $\mathcal{L}_N$ that is not global minimum can exist.

Since the activation function $\sigma$ is piecewise linear, the entire parameter space can be partitioned into disjoint cones. In particular, we consider the set of cones $\mathcal{K}_{I \times J}$ where $I \subset \{1, \ldots, d_1\}, J \subset \{1, \ldots, m\}$ that satisfy

$$\mathcal{K}_{I \times J} := \{(\boldsymbol{A}_2, \boldsymbol{A}_1) : (\boldsymbol{A}_1)_{I,:} \boldsymbol{X}_{:,J} \geq 0, \text{other entries of } \boldsymbol{A}_1 \boldsymbol{X} < 0\}, \tag{8}$$

where "$\geq$" and "$<$" represent entrywise comparisons. Within $\mathcal{K}_{I \times J}$, the term $\sigma(\boldsymbol{A}_1 \boldsymbol{X})$ activates only the entries $\sigma(\boldsymbol{A}_1 \boldsymbol{X})_{I:J}$, and the corresponding loss function $\mathcal{L}_N$ is equivalent to

$$\forall (\boldsymbol{A}_2, \boldsymbol{A}_1) \in \mathcal{K}_{I \times J}, \quad \mathcal{L}_N := \tfrac{1}{2} \left\| (\boldsymbol{A}_2)_{:,I} (\boldsymbol{A}_1)_{I,:} \boldsymbol{X}_{:,J} - \boldsymbol{Y}_{:,J} \right\|_F^2 + \tfrac{1}{2} \left\| \boldsymbol{Y}_{:,J^c} \right\|_F^2. \tag{9}$$

Hence, within $\mathcal{K}_{I \times J}$, $\mathcal{L}_N$ reduces to the loss of a shallow linear network with parameters $((\boldsymbol{A}_2)_{:,I}, (\boldsymbol{A}_1)_{I,:})$ and input & output data pair $(\boldsymbol{X}_{:,J}, \boldsymbol{Y}_{:,J})$. Note that our results on shallow linear networks in Section 2 are applicable to all parameter dimensions and data matrices. Thus, Theorem 1 fully characterizes the forms of critical points of $\mathcal{L}_N$ in $\mathcal{K}_{I \times J}$. Moreover, the existence of such critical points can be analytically examined by substituting their forms into eq. (8). In summary, we obtain the following result, where we denote $\boldsymbol{\Sigma}_J := \boldsymbol{Y}_{:,J} \boldsymbol{X}_{:,J}^\dagger \boldsymbol{X}_{:,J} \boldsymbol{Y}_{:,J}^\top$ with the full singular value decomposition $\boldsymbol{U}_J \boldsymbol{\Lambda}_J \boldsymbol{U}_J^\top$, and suppose that $\boldsymbol{\Sigma}_J$ has $r(J)$ distinct positive singular values $\sigma_1(J) > \cdots > \sigma_{r(J)}(J)$ with multiplicities $m_1, \ldots, m_{r(J)}$, respectively, and $\bar{m}(J)$ zero singular values.

**Proposition 6** (Characterization of critical points). *All critical points of $\mathcal{L}_N$ in $\mathcal{K}_{I \times J}$ for any $I \subset \{1, \ldots, d_1\}, J \subset \{1, \ldots, m\}$ are necessarily and sufficiently characterized by an $\boldsymbol{L}_1 \in \mathbb{R}^{|I| \times d_0}$, a block matrix $\boldsymbol{V} \in \mathbb{R}^{d_2 \times |I|}$ and an invertible matrix $\boldsymbol{C} \in \mathbb{R}^{|I| \times |I|}$ such that*

$$(\boldsymbol{A}_1)_{I,:} = \boldsymbol{C}^{-1} \boldsymbol{V}^\top \boldsymbol{U}_J^\top \boldsymbol{Y}_{:,J} \boldsymbol{X}_{:,J}^\dagger + \boldsymbol{L}_1 - \boldsymbol{C}^{-1} \boldsymbol{V}^\top \boldsymbol{V} \boldsymbol{C} \boldsymbol{L}_1 \boldsymbol{X}_{:,J} \boldsymbol{X}_{:,J}^\dagger, \tag{10}$$

$$(\boldsymbol{A}_2)_{:,I} = \boldsymbol{U}_J \boldsymbol{V} \boldsymbol{C}. \tag{11}$$

*Specifically, $\boldsymbol{V} = [\text{diag}(\boldsymbol{V}_1, \ldots, \boldsymbol{V}_{r(J)}, \overline{\boldsymbol{V}}), \boldsymbol{0}^{d_2 \times (|I| - \text{rank}((\boldsymbol{A}_2)_{:,I}))}]$, where $\boldsymbol{V}_i \in \mathbb{R}^{m_i \times p_i}, \overline{\boldsymbol{V}} \in \mathbb{R}^{\bar{m} \times \bar{p}}$ consist of orthonormal columns with $p_i \leq m_i$ for $i = 1, \ldots, r(J), \bar{p} \leq \bar{m}$ such that $\sum_{i=1}^{r(J)} p_i + \bar{p} = \text{rank}((\boldsymbol{A}_2)_{:,I})$, and $\boldsymbol{L}_1, \boldsymbol{V}, \boldsymbol{C}$ satisfy*

$$\mathcal{P}_{\text{col}(\boldsymbol{U}_J \boldsymbol{V})^\perp} \boldsymbol{Y}_{:,J} \boldsymbol{X}_{:,J}^\top \boldsymbol{L}_1^\top \boldsymbol{C}^\top \mathcal{P}_{\text{ker}(\boldsymbol{V})} = \boldsymbol{0}. \tag{12}$$

*Moreover, a critical point in $\mathcal{K}_{I \times J}$ exists if and only if there exists such $\boldsymbol{C}, \boldsymbol{V}, \boldsymbol{L}_1$ that*

$$(\boldsymbol{A}_1)_{I,:} \boldsymbol{X}_{:,J} = \boldsymbol{C}^{-1} \boldsymbol{V}^\top \boldsymbol{U}_J^\top \boldsymbol{Y} \boldsymbol{X}_{:,J}^\dagger \boldsymbol{X}_{:,J} + \boldsymbol{C}^{-1} (\boldsymbol{I} - \mathcal{P}_{\text{ker}(\boldsymbol{V})}) \boldsymbol{C} \boldsymbol{L}_1 \boldsymbol{X}_{:,J} \geq 0, \tag{13}$$

$$\text{Other entries of } \boldsymbol{A}_1 \boldsymbol{X} < 0. \tag{14}$$

To further illustrate, we consider a special case where the nonlinear network has one unit in the hidden layer, i.e., $d_1 = 1$, in which case $\boldsymbol{A}_1$ and $\boldsymbol{A}_2$ are row and column vectors, respectively. Then, the entire parameter space can be partitioned into disjoint cones taking the form of $\mathcal{K}_{I \times J}$, and $I = \{1\}$ is the only nontrivial choice. We obtain the following result from Proposition 6.

**Proposition 7** (Characterization of critical points). *Consider $\mathcal{L}_N$ with $d_1 = 1$ and any $J \subset \{1, \ldots, m\}$. Then, any nonzero critical point of $\mathcal{L}_N$ within $\mathcal{K}_{\{1\} \times J}$ can be necessarily and sufficiently characterized by an $\boldsymbol{\ell}_1^\top \in \mathbb{R}^{1 \times d_0}$, a block unit vector $\mathbf{v} \in \mathbb{R}^{d_2 \times 1}$ and a scalar $c \in \mathbb{R}$ such that*

$$\boldsymbol{A}_1 = c^{-1} \mathbf{v}^\top \boldsymbol{U}_J^\top \boldsymbol{Y}_{:,J} \boldsymbol{X}_{:,J}^\dagger + \boldsymbol{\ell}_1^\top - \boldsymbol{\ell}_1^\top \boldsymbol{X}_{:,J} \boldsymbol{X}_{:,J}^\dagger, \quad \boldsymbol{A}_2 = c \boldsymbol{U}_J \mathbf{v}. \tag{15}$$

*Specifically, $\mathbf{v}$ is a unit vector that is supported on the entries corresponding to the same singular value of $\boldsymbol{\Sigma}_J$. Moreover, a nonzero critical point in $\mathcal{K}_{\{1\} \times J}$ exists if and only if there exist such $c, \mathbf{v}, \boldsymbol{\ell}_1^\top$ that satisfy*

$$\boldsymbol{A}_1 \boldsymbol{X}_{:,J} = c^{-1} \mathbf{v}^\top \boldsymbol{U}_J^\top \boldsymbol{Y}_{:,J} \boldsymbol{X}_{:,J}^\dagger \boldsymbol{X}_{:,J} \geq 0, \tag{16}$$

$$\boldsymbol{A}_1 \boldsymbol{X}_{:,J^c} = c^{-1} \mathbf{v}^\top \boldsymbol{U}_J^\top \boldsymbol{Y}_{:,J} \boldsymbol{X}_{:,J}^\dagger \boldsymbol{X}_{:,J^c} + \boldsymbol{\ell}_1^\top \boldsymbol{X}_{:,J^c} - \boldsymbol{\ell}_1^\top \boldsymbol{X}_{:,J} \boldsymbol{X}_{:,J}^\dagger \boldsymbol{X}_{:,J^c} < 0. \tag{17}$$

We note that Proposition 7 characterizes both the existence and the forms of critical points of $\mathcal{L}_N$ over the entire parameter space for nonlinear networks with a single hidden unit. The condition in eq. (12) is guaranteed because $\mathcal{P}_{\ker(\mathbf{v})} = \mathbf{0}$ for $\mathbf{v} \neq \mathbf{0}$.

To further understand Proposition 7, suppose that there exists a critical point in $\mathcal{K}_{\{1\} \times J}$ with $\mathbf{v}$ being supported on the entries that correspond to the $i$-th singular value of $\boldsymbol{\Sigma}_J$. Then, Proposition 1 implies that $\mathcal{L}_N = \frac{1}{2} \text{Tr}(\boldsymbol{Y} \boldsymbol{Y}^\top) - \frac{1}{2} \sigma_i(J)$. In particular, the critical point achieves the local minimum $\frac{1}{2} \text{Tr}(\boldsymbol{Y} \boldsymbol{Y}^\top) - \frac{1}{2} \sigma_1(J)$ in $\mathcal{K}_{\{1\} \times J}$ with $i = 1$. This is because in this case the critical point is full rank with an optimal order, and hence corresponds to the global minimum of the linear network in eq. (9). Since the singular values of $\boldsymbol{\Sigma}_J$ may vary with the choice of $J$, $\mathcal{L}_N$ may achieve different local minima in different cones. Thus, local minimum that is not global minimum can exist for $\mathcal{L}_N$. The following proposition concludes this fact by considering a concrete example.

**Proposition 8.** *For one-hidden-layer nonlinear neural networks with ReLU activation function, there exists local minimum that is not global minimum, and there also exists local maximum.*

The above proposition is demonstrated by the following example.

**Example 1.** *Consider the loss function $\mathcal{L}_N$ of the nonlinear network with $d_2 = d_0 = 2$, and $d_1 = 1$. The input and output data are set to be $\boldsymbol{X} = \text{diag}(1, 1), \boldsymbol{Y} = \text{diag}(2, 1)$.*

First, consider the cone $\mathcal{K}_{I \times J}$ with $I = \{1\}, J = \{1\}$. Calculation yields that $\boldsymbol{\Sigma}_J = \text{diag}(4, 0)$, and the conditions for existence of critical points in eqs. (16) and (17) hold if $2c^{-1}(\mathbf{v})_{1,:} \geq 0, (\boldsymbol{\ell}_1)_{2,:} < 0$. Then choosing $c = 1, \mathbf{v} = (1, 0)^\top, \boldsymbol{\ell}_1 = (1, -1)^\top$ yields a local minimum in $\mathcal{K}_{I \times J}$, because the nonzero entry in $\mathbf{v}$ corresponds to the largest singular value of $\boldsymbol{\Sigma}_J$. Then, calculation shows that the local minimum achieves $\mathcal{L}_N = \frac{1}{2}$. On the other hand, consider the cone $\mathcal{K}_{I \times J'}$ with $I = \{1\}, J' = \{2\}$, in which $\boldsymbol{\Sigma}_{J'} = \text{diag}(0, 1)$. The conditions for existence of critical points in eqs. (16) and (17) hold if $c^{-1}(\mathbf{v})_{1,:} \geq 0, (\boldsymbol{\ell}_1)_{1,:} < 0$. Similarly to the previous case, choosing $c = 1, \mathbf{v} = (1, 0)^\top, \boldsymbol{\ell}_1 = (-1, 0)^\top$ yields a local minimum that achieves the function value $\mathcal{L}_n = 2$. Hence, local minimum that is not global minimum does exist. Moreover, in the cone $\mathcal{K}_{I \times J''}$ with $I = \{1\}, J'' = \emptyset$, the function $\mathcal{L}_N$ remains to be the constant $\frac{5}{2}$, and all points in this cone are local minimum or local maximum. Thus, the landscape of the loss function of nonlinear networks is very different from that of the loss function of linear networks.

## CONCLUSION

In this paper, we provide full characterization of the analytical forms of the critical points for the square loss function of three types of neural networks, namely, shallow linear networks, deep linear networks, and shallow ReLU nonlinear networks. We show that such analytical forms of the critical points have direct implications on the values of the corresponding loss functions, achievement of global minimum, and various landscape properties around these critical points. As a consequence, the loss function for linear networks has no spurious local minimum, while such point does exist for nonlinear networks with ReLU activation. In the future, it is interesting to further explore nonlinear neural networks. In particular, we wish to characterize the analytical form of critical points for deep nonlinear networks and over the full parameter space. Such results will further facilitate the understanding of the landscape properties around these critical points.

ACKNOWLEDGMENTS

The work was supported in part by US National Science Foundation under the grants CCF-1761506, ECCS-1818904, and was supported in part by DARPA FunLoL program.

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

# Supplementary Materials

PROOF OF THEOREM 1

Notations: For any matrix $\boldsymbol{M}$, denote $\mathrm{vec}(\boldsymbol{M})$ as the column vector formed by stacking its columns. Denote the Kronecker product as "$\otimes$". Then, the following useful relationships hold for any dimension compatible matrices $\boldsymbol{M}, \boldsymbol{U}, \boldsymbol{V}, \boldsymbol{W}$:

$$\mathrm{vec}(\boldsymbol{U}\boldsymbol{M}\boldsymbol{V}) = (\boldsymbol{V}^\top \otimes \boldsymbol{U})\mathrm{vec}(\boldsymbol{M}), \tag{18}$$

$$(\boldsymbol{U} \otimes \boldsymbol{V})^\dagger = \boldsymbol{U}^\dagger \otimes \boldsymbol{V}^\dagger, \tag{19}$$

$$(\boldsymbol{M} \otimes \boldsymbol{W})(\boldsymbol{U} \otimes \boldsymbol{V}) = (\boldsymbol{M}\boldsymbol{U}) \otimes (\boldsymbol{W}\boldsymbol{V}), \tag{20}$$

$$(\boldsymbol{M}^\top \boldsymbol{M})^\dagger \boldsymbol{M}^\top = \boldsymbol{M}^\top (\boldsymbol{M}\boldsymbol{M}^\top)^\dagger = \boldsymbol{M}^\dagger, \tag{21}$$

$$\boldsymbol{M}\boldsymbol{M}^\dagger \boldsymbol{M} = \boldsymbol{M}, \; \boldsymbol{M}^\dagger \boldsymbol{M}\boldsymbol{M}^\dagger = \boldsymbol{M}^\dagger. \tag{22}$$

Recall that a point $(\boldsymbol{A}_1, \boldsymbol{A}_2)$ is a critical point of $\mathcal{L}$ if it satisfies

$$\nabla_{\boldsymbol{A}_1}\mathcal{L} = \boldsymbol{A}_2^\top (\boldsymbol{A}_2\boldsymbol{A}_1\boldsymbol{X} - \boldsymbol{Y})\boldsymbol{X}^\top = \boldsymbol{0}, \tag{23}$$

$$\nabla_{\boldsymbol{A}_2}\mathcal{L} = (\boldsymbol{A}_2\boldsymbol{A}_1\boldsymbol{X} - \boldsymbol{Y})\boldsymbol{X}^\top \boldsymbol{A}_1^\top = \boldsymbol{0}. \tag{24}$$

We first prove eqs. (1) and (2).

**Lemma 1.** *Let $(\boldsymbol{A}_2, \boldsymbol{A}_1)$ be a critical point of $\mathcal{L}$. Then it must satisfy, for some $\boldsymbol{L}_1 \in \mathbb{R}^{d_1 \times d_0}$, that*

$$\boldsymbol{A}_1 = \boldsymbol{A}_2^\dagger \boldsymbol{Y}\boldsymbol{X}^\dagger + \boldsymbol{L}_1 - \boldsymbol{A}_2^\dagger \boldsymbol{A}_2\boldsymbol{L}_1\boldsymbol{X}\boldsymbol{X}^\dagger, \tag{25}$$

$$\mathcal{P}_{\mathrm{col}(\boldsymbol{A}_2)}\boldsymbol{\Sigma}\,\mathcal{P}_{\mathrm{col}(\boldsymbol{A}_2)} = \boldsymbol{\Sigma}\,\mathcal{P}_{\mathrm{col}(\boldsymbol{A}_2)} = \mathcal{P}_{\mathrm{col}(\boldsymbol{A}_2)}\boldsymbol{\Sigma}. \tag{26}$$

*Proof of Lemma 1.* Since $(\boldsymbol{A}_2, \boldsymbol{A}_1)$ is a critical point of $\mathcal{L}$, eq. (23) implies that

$$\boldsymbol{A}_2^\top \boldsymbol{A}_2\boldsymbol{A}_1\boldsymbol{X}\boldsymbol{X}^\top = \boldsymbol{A}_2^\top \boldsymbol{Y}\boldsymbol{X}^\top.$$

Applying the vectorizing operator on both sides of the above equation and use the property in eq. (18), we conclude that

$$(\boldsymbol{X}\boldsymbol{X}^\top \otimes \boldsymbol{A}_2^\top \boldsymbol{A}_2)\mathrm{vec}(\boldsymbol{A}_1) = (\boldsymbol{X} \otimes \boldsymbol{A}_2^\top)\mathrm{vec}(\boldsymbol{Y}).$$

Since $\mathrm{vec}(\boldsymbol{A}_1)$ is a solution of the above linear equation, it must take the form of the solution of linear systems, i.e., for some $\boldsymbol{L}_1 \in \mathbb{R}^{d_1 \times d_0}$, we have

$$\mathrm{vec}(\boldsymbol{A}_1) = (\boldsymbol{X}\boldsymbol{X}^\top \otimes \boldsymbol{A}_2^\top \boldsymbol{A}_2)^\dagger (\boldsymbol{X} \otimes \boldsymbol{A}_2^\top)\mathrm{vec}(\boldsymbol{Y}) + [\boldsymbol{I} - (\boldsymbol{X}\boldsymbol{X}^\top \otimes \boldsymbol{A}_2^\top \boldsymbol{A}_2)^\dagger (\boldsymbol{X}\boldsymbol{X}^\top \otimes \boldsymbol{A}_2^\top \boldsymbol{A}_2)]\mathrm{vec}(\boldsymbol{L}_1)$$

$$\overset{(i)}{=} \left((\boldsymbol{X}\boldsymbol{X}^\top)^\dagger \boldsymbol{X} \otimes (\boldsymbol{A}_2^\top \boldsymbol{A}_2)^\dagger \boldsymbol{A}_2^\top\right)\mathrm{vec}(\boldsymbol{Y}) + [\boldsymbol{I} - (\boldsymbol{X}\boldsymbol{X}^\top)^\dagger \boldsymbol{X}\boldsymbol{X}^\top \otimes (\boldsymbol{A}_2^\top \boldsymbol{A}_2)^\dagger \boldsymbol{A}_2^\top \boldsymbol{A}_2]\mathrm{vec}(\boldsymbol{L}_1)$$

$$= \mathrm{vec}\left((\boldsymbol{A}_2^\top \boldsymbol{A}_2)^\dagger \boldsymbol{A}_2^\top \boldsymbol{Y}\boldsymbol{X}^\top (\boldsymbol{X}\boldsymbol{X}^\top)^\dagger + \boldsymbol{L}_1 - (\boldsymbol{A}_2^\top \boldsymbol{A}_2)^\dagger \boldsymbol{A}_2^\top \boldsymbol{A}_2\boldsymbol{L}_1\boldsymbol{X}\boldsymbol{X}^\top (\boldsymbol{X}\boldsymbol{X}^\top)^\dagger\right)$$

$$\overset{(ii)}{=} \mathrm{vec}\left(\boldsymbol{A}_2^\dagger \boldsymbol{Y}\boldsymbol{X}^\dagger + \boldsymbol{L}_1 - \boldsymbol{A}_2^\dagger \boldsymbol{A}_2\boldsymbol{L}_1\boldsymbol{X}\boldsymbol{X}^\dagger\right)$$

where (i) uses eqs. (19) and (20) and (ii) uses eq. (21). Then, eq. (25) follows by reshaping the vector into a matrix.

Next we prove eq. (26). Multiplying both sides of eq. (25) by $\boldsymbol{A}_2$ on the left and by $\boldsymbol{X}$ on the right and then using eq. (22), we obtain

$$\boldsymbol{A}_2\boldsymbol{A}_1\boldsymbol{X} = \boldsymbol{A}_2\boldsymbol{A}_2^\dagger \boldsymbol{Y}\boldsymbol{X}^\dagger \boldsymbol{X} = \mathcal{P}_{\mathrm{col}(\boldsymbol{A}_2)}\boldsymbol{Y}\boldsymbol{X}^\dagger \boldsymbol{X}. \tag{27}$$

Also, multiplying both sides of eq. (24) by $\boldsymbol{A}_2^\top$ on the right yields that $\boldsymbol{A}_2\boldsymbol{A}_1\boldsymbol{X}\boldsymbol{X}^\top \boldsymbol{A}_1^\top \boldsymbol{A}_2^\top = \boldsymbol{Y}\boldsymbol{X}^\top \boldsymbol{A}_1^\top \boldsymbol{A}_2^\top$. This equation, together with the above expression of $\boldsymbol{A}_2\boldsymbol{A}_1\boldsymbol{X}$, further implies that

$$\mathcal{P}_{\mathrm{col}(\boldsymbol{A}_2)}\boldsymbol{\Sigma}\,\mathcal{P}_{\mathrm{col}(\boldsymbol{A}_2)} = \boldsymbol{\Sigma}\,\mathcal{P}_{\mathrm{col}(\boldsymbol{A}_2)}.$$

Note that $\mathcal{P}_{\mathrm{col}(\boldsymbol{A}_2)}\boldsymbol{\Sigma}\,\mathcal{P}_{\mathrm{col}(\boldsymbol{A}_2)}$ is symmetric. Thus, we conclude that $\boldsymbol{\Sigma}\,\mathcal{P}_{\mathrm{col}(\boldsymbol{A}_2)} = \mathcal{P}_{\mathrm{col}(\boldsymbol{A}_2)}\boldsymbol{\Sigma}$. $\square$

Next, we derive the form of $\boldsymbol{A}_2$. Recall the full singular value decomposition $\boldsymbol{\Sigma} = \boldsymbol{U}\boldsymbol{\Lambda}\boldsymbol{U}^\top$, where $\boldsymbol{\Lambda}$ is a diagonal matrix with distinct singular values $\sigma_1 > \ldots > \sigma_r > 0$ and multiplicities $m_1, \ldots, m_r$, respectively. We also assume that there are $\bar{m}$ number of zero singular values in $\boldsymbol{\Lambda}$. Using the fact that $\mathcal{P}_{\mathrm{col}(\boldsymbol{A}_2)} = \boldsymbol{U}\,\mathcal{P}_{\mathrm{col}(\boldsymbol{U}^\top \boldsymbol{A}_2)}\boldsymbol{U}^\top$, the last equality in eq. (26) reduces to

$$\boldsymbol{\Lambda}\,\mathcal{P}_{\mathrm{col}(\boldsymbol{U}^\top\boldsymbol{A}_2)} = \mathcal{P}_{\mathrm{col}(\boldsymbol{U}^\top\boldsymbol{A}_2)}\boldsymbol{\Lambda}.$$

By the multiplicity pattern of the singular values in $\boldsymbol{\Lambda}$, $\mathcal{P}_{\mathrm{col}(\boldsymbol{U}^\top\boldsymbol{A}_2)}$ must be block diagonal. Specifically, we can write $\mathcal{P}_{\mathrm{col}(\boldsymbol{U}^\top\boldsymbol{A}_2)} = \mathrm{diag}(\mathcal{P}_1, \ldots, \mathcal{P}_r, \overline{\mathcal{P}})$, where $\mathcal{P}_i \in \mathbb{R}^{m_i \times m_i}$ and $\overline{\mathcal{P}} \in \mathbb{R}^{\bar{m} \times \bar{m}}$. Also, since $\mathcal{P}_{\mathrm{col}(\boldsymbol{U}^\top\boldsymbol{A}_2)}$ is a projection, $\mathcal{P}_1, \ldots, \mathcal{P}_r, \overline{\mathcal{P}}$ must all be projections. Note that $\mathcal{P}_{\mathrm{col}(\boldsymbol{U}^\top\boldsymbol{A}_2)}$ has rank $\mathrm{rank}(\boldsymbol{A}_2)$, and suppose that $\mathcal{P}_1, \ldots, \mathcal{P}_r, \overline{\mathcal{P}}$ have ranks $p_1, \ldots, p_r, \bar{p}$, respectively. Then, we must have $p_i \leq m_i$ for $i = 1, \ldots, r$, $\bar{p} \leq \bar{m}$ and $\sum_{i=1}^r p_i + \bar{p} = \mathrm{rank}(\boldsymbol{A}_2)$. Also, note that each projection can be expressed as $\mathcal{P}_i = \boldsymbol{V}_i\boldsymbol{V}_i^\top$ with $\boldsymbol{V}_i \in \mathbb{R}^{m_i \times p_i}$, $\overline{\boldsymbol{V}} \in \mathbb{R}^{\bar{m} \times \bar{p}}$ consisting of orthonormal columns. Hence, we can write $\mathcal{P}_{\mathrm{col}(\boldsymbol{U}^\top\boldsymbol{A}_2)} = \widehat{\boldsymbol{V}}\widehat{\boldsymbol{V}}^\top$ where $\widehat{\boldsymbol{V}} = \mathrm{diag}(\boldsymbol{V}_1, \ldots, \boldsymbol{V}_r, \overline{\boldsymbol{V}})$. We then conclude that $\mathcal{P}_{\mathrm{col}(\boldsymbol{A}_2)} = \boldsymbol{U}\,\mathcal{P}_{\mathrm{col}(\boldsymbol{U}^\top\boldsymbol{A}_2)}\boldsymbol{U}^\top = \boldsymbol{U}\widehat{\boldsymbol{V}}\widehat{\boldsymbol{V}}^\top\boldsymbol{U}^\top$. Thus, $\boldsymbol{A}_2$ has the same column space as $\boldsymbol{U}\widehat{\boldsymbol{V}}$, and there must exist an invertible matrix $\boldsymbol{C} \in \mathbb{R}^{d_1 \times d_1}$ such that $\boldsymbol{A}_2 = \boldsymbol{U}[\widehat{\boldsymbol{V}}, \boldsymbol{0}]\boldsymbol{C}$, where $\boldsymbol{0} \in \mathbb{R}^{d_2 \times (d_1 - \mathrm{rank}(\boldsymbol{A}_2))}$ is a zero matrix. Denoting $\boldsymbol{V} = [\widehat{\boldsymbol{V}}, \boldsymbol{0}]$, we conclude that $\boldsymbol{A}_2 = \boldsymbol{U}\boldsymbol{V}\boldsymbol{C}$. Then, plugging $\boldsymbol{A}_2^\dagger = \boldsymbol{C}^{-1}\boldsymbol{V}^\top\boldsymbol{U}^\top$ into eq. (25) yields the desired form of $\boldsymbol{A}_1$.

We now prove eq. (3). Note that the above proof is based on the equations $\nabla_{\boldsymbol{A}_1}\mathcal{L} = \boldsymbol{0}$, $(\nabla_{\boldsymbol{A}_2}\mathcal{L})\boldsymbol{A}_2^\top = \boldsymbol{0}$. Hence, the forms of $\boldsymbol{A}_1, \boldsymbol{A}_2$ in eqs. (1) and (2) need to further satisfy $\nabla_{\boldsymbol{A}_2}\mathcal{L} = \boldsymbol{0}$. By eq. (27) and the form of $\boldsymbol{A}_2$, we obtain that $\boldsymbol{A}_2\boldsymbol{A}_1\boldsymbol{X} = \mathcal{P}_{\mathrm{col}(\boldsymbol{A}_2)}\boldsymbol{Y}\boldsymbol{X}^\dagger\boldsymbol{X} = \boldsymbol{U}\boldsymbol{V}(\boldsymbol{U}\boldsymbol{V})^\top\boldsymbol{Y}\boldsymbol{X}^\dagger\boldsymbol{X}$. This expression, together with the form of $\boldsymbol{A}_1$ in eq. (1), implies that

$$\begin{aligned}
\boldsymbol{A}_2\boldsymbol{A}_1\boldsymbol{X}\boldsymbol{X}^\top\boldsymbol{A}_1^\top &= \boldsymbol{U}\boldsymbol{V}(\boldsymbol{U}\boldsymbol{V})^\top\boldsymbol{Y}\boldsymbol{X}^\dagger\boldsymbol{X}\boldsymbol{X}^\top\boldsymbol{A}_1^\top \\
&\overset{(i)}{=} \boldsymbol{U}\boldsymbol{V}(\boldsymbol{U}\boldsymbol{V})^\top\boldsymbol{Y}\boldsymbol{X}^\top\boldsymbol{A}_1^\top \\
&= \boldsymbol{U}\boldsymbol{V}(\boldsymbol{U}\boldsymbol{V})^\top\boldsymbol{\Sigma}\boldsymbol{U}\boldsymbol{V}(\boldsymbol{C}^\top)^{-1} + \boldsymbol{U}\boldsymbol{V}(\boldsymbol{U}\boldsymbol{V})^\top\boldsymbol{Y}\boldsymbol{X}^\top\boldsymbol{L}_1^\top \\
&\quad - \boldsymbol{U}\boldsymbol{V}(\boldsymbol{U}\boldsymbol{V})^\top\boldsymbol{Y}\boldsymbol{X}^\top\boldsymbol{L}_1^\top\boldsymbol{C}^\top\boldsymbol{V}^\top\boldsymbol{V}(\boldsymbol{C}^\top)^{-1} \\
&= \boldsymbol{U}\boldsymbol{V}\boldsymbol{V}^\top\boldsymbol{\Lambda}\boldsymbol{V}(\boldsymbol{C}^\top)^{-1} + \boldsymbol{U}\boldsymbol{V}(\boldsymbol{U}\boldsymbol{V})^\top\boldsymbol{Y}\boldsymbol{X}^\top\boldsymbol{L}_1^\top(\boldsymbol{I} - \boldsymbol{C}^\top\boldsymbol{V}^\top\boldsymbol{V}(\boldsymbol{C}^\top)^{-1}) \\
&\overset{(ii)}{=} \boldsymbol{U}\boldsymbol{\Lambda}\boldsymbol{V}(\boldsymbol{C}^\top)^{-1} + \boldsymbol{U}\boldsymbol{V}(\boldsymbol{U}\boldsymbol{V})^\top\boldsymbol{Y}\boldsymbol{X}^\top\boldsymbol{L}_1^\top(\boldsymbol{I} - \boldsymbol{C}^\top\boldsymbol{V}^\top\boldsymbol{V}(\boldsymbol{C}^\top)^{-1}),
\end{aligned}$$

where (i) uses the fact that $\boldsymbol{X}^\dagger\boldsymbol{X}\boldsymbol{X}^\top = \boldsymbol{X}^\top$, (ii) uses the fact that the block pattern of $\boldsymbol{V}$ is compatible with the multiplicity pattern of the singular values in $\boldsymbol{\Lambda}$, and hence $\boldsymbol{V}\boldsymbol{V}^\top\boldsymbol{\Lambda}\boldsymbol{V} = \boldsymbol{\Lambda}\boldsymbol{V}$. On the other hand, we also obtain that

$$\begin{aligned}
\boldsymbol{Y}\boldsymbol{X}^\top\boldsymbol{A}_1^\top &= \boldsymbol{\Sigma}\boldsymbol{U}\boldsymbol{V}(\boldsymbol{C}^\top)^{-1} + \boldsymbol{Y}\boldsymbol{X}^\top\boldsymbol{L}_1^\top(\boldsymbol{I} - \boldsymbol{C}^\top\boldsymbol{V}^\top\boldsymbol{V}(\boldsymbol{C}^\top)^{-1}) \\
&= \boldsymbol{U}\boldsymbol{\Lambda}\boldsymbol{V}(\boldsymbol{C}^\top)^{-1} + \boldsymbol{Y}\boldsymbol{X}^\top\boldsymbol{L}_1^\top(\boldsymbol{I} - \boldsymbol{C}^\top\boldsymbol{V}^\top\boldsymbol{V}(\boldsymbol{C}^\top)^{-1}).
\end{aligned}$$

Thus, to satisfy $\nabla_{\boldsymbol{A}_2}\mathcal{L} = \boldsymbol{0}$ in eq. (24), we require that

$$(\boldsymbol{I} - \boldsymbol{U}\boldsymbol{V}(\boldsymbol{U}\boldsymbol{V})^\top)\boldsymbol{Y}\boldsymbol{X}^\top\boldsymbol{L}_1^\top(\boldsymbol{I} - \boldsymbol{C}^\top\boldsymbol{V}^\top\boldsymbol{V}(\boldsymbol{C}^\top)^{-1}) = \boldsymbol{0},$$

which is equivalent to

$$(\boldsymbol{I} - \boldsymbol{U}\boldsymbol{V}(\boldsymbol{U}\boldsymbol{V})^\top)\boldsymbol{Y}\boldsymbol{X}^\top\boldsymbol{L}_1^\top\boldsymbol{C}^\top(\boldsymbol{I} - \boldsymbol{V}^\top\boldsymbol{V}) = \boldsymbol{0}.$$

Lastly, note that $(\boldsymbol{I} - \boldsymbol{U}\boldsymbol{V}(\boldsymbol{U}\boldsymbol{V})^\top) = \mathcal{P}_{\mathrm{col}(\boldsymbol{U}\boldsymbol{V})^\perp}$, and $(\boldsymbol{I} - \boldsymbol{V}^\top\boldsymbol{V}) = \mathcal{P}_{\mathrm{ker}(\boldsymbol{V})}$, which concludes the proof.

## PROOF OF PROPOSITION 1

By expansion we obtain that $\mathcal{L} = \frac{1}{2}\mathrm{Tr}(\boldsymbol{Y}\boldsymbol{Y}^\top) - \mathrm{Tr}(\boldsymbol{A}_2\boldsymbol{A}_1\boldsymbol{X}\boldsymbol{Y}^\top) + \frac{1}{2}\mathrm{Tr}(\boldsymbol{A}_2\boldsymbol{A}_1\boldsymbol{X}\boldsymbol{X}^\top\boldsymbol{A}_1^\top\boldsymbol{A}_2^\top)$. Consider any $(\boldsymbol{A}_1, \boldsymbol{A}_2)$ that satisfies eq. (23), we have shown that such a point also satisfies eq. (27),

which further yields that

$$\mathcal{L} = \tfrac{1}{2}\mathrm{Tr}(\boldsymbol{Y}\boldsymbol{Y}^\top) - \mathrm{Tr}(\boldsymbol{A}_2\boldsymbol{A}_1\boldsymbol{X}\boldsymbol{Y}^\top) + \tfrac{1}{2}\mathrm{Tr}(\boldsymbol{A}_2\boldsymbol{A}_1\boldsymbol{X}\boldsymbol{X}^\top\boldsymbol{A}_1^\top\boldsymbol{A}_2^\top)$$

$$= \tfrac{1}{2}\mathrm{Tr}(\boldsymbol{Y}\boldsymbol{Y}^\top) - \mathrm{Tr}(\mathcal{P}_{\mathrm{col}(\boldsymbol{A}_2)}\boldsymbol{\Sigma}) + \tfrac{1}{2}\mathrm{Tr}(\mathcal{P}_{\mathrm{col}(\boldsymbol{A}_2)}\boldsymbol{\Sigma}\,\mathcal{P}_{\mathrm{col}(\boldsymbol{A}_2)})$$

$$\overset{(i)}{=} \tfrac{1}{2}\mathrm{Tr}(\boldsymbol{Y}\boldsymbol{Y}^\top) - \tfrac{1}{2}\mathrm{Tr}(\mathcal{P}_{\mathrm{col}(\boldsymbol{A}_2)}\boldsymbol{\Sigma})$$

$$\overset{(ii)}{=} \tfrac{1}{2}\mathrm{Tr}(\boldsymbol{Y}\boldsymbol{Y}^\top) - \tfrac{1}{2}\mathrm{Tr}(\mathcal{P}_{\mathrm{col}(\boldsymbol{U}^\top\boldsymbol{A}_2)}\boldsymbol{\Lambda}) \tag{28}$$

where (i) follows from the fact that $\mathrm{Tr}(\mathcal{P}_{\mathrm{col}(\boldsymbol{A}_2)}\boldsymbol{\Sigma}\,\mathcal{P}_{\mathrm{col}(\boldsymbol{A}_2)}) = \mathrm{Tr}(\mathcal{P}_{\mathrm{col}(\boldsymbol{A}_2)}\boldsymbol{\Sigma})$, and (ii) uses the fact that $\mathcal{P}_{\mathrm{col}(\boldsymbol{A}_2)} = \boldsymbol{U}\,\mathcal{P}_{\mathrm{col}(\boldsymbol{U}^\top\boldsymbol{A}_2)}\boldsymbol{U}^\top$. In particular, a critical point $(\boldsymbol{A}_1,\boldsymbol{A}_2)$ satisfies eq. (28). Moreover, using the form of the critical point $\boldsymbol{A}_2 = \boldsymbol{U}\boldsymbol{V}\boldsymbol{C}$, eq. (28) further becomes

$$\mathcal{L} = \tfrac{1}{2}\mathrm{Tr}(\boldsymbol{Y}\boldsymbol{Y}^\top) - \tfrac{1}{2}\mathrm{Tr}(\mathcal{P}_{\mathrm{col}(\boldsymbol{V}\boldsymbol{C})}\boldsymbol{\Lambda})$$

$$\overset{(i)}{=} \tfrac{1}{2}\mathrm{Tr}(\boldsymbol{Y}\boldsymbol{Y}^\top) - \tfrac{1}{2}\mathrm{Tr}(\boldsymbol{V}^\top\boldsymbol{\Lambda}\boldsymbol{V})$$

$$\overset{(ii)}{=} \tfrac{1}{2}\mathrm{Tr}(\boldsymbol{Y}\boldsymbol{Y}^\top) - \tfrac{1}{2}\sum_{i=1}^{r} p_i\sigma_i,$$

where (i) is due to $\mathcal{P}_{\mathrm{col}(\boldsymbol{V}\boldsymbol{C})} = \mathcal{P}_{\mathrm{col}(\boldsymbol{V})} = \boldsymbol{V}\boldsymbol{V}^\top$, and (ii) utilizes the block pattern of $\boldsymbol{V}$ and the multiplicity pattern of $\boldsymbol{\Lambda}$ that are specified in Theorem 1.

## PROOF OF PROPOSITION 2

(1): Consider a critical point $(\boldsymbol{A}_1,\boldsymbol{A}_2)$ with the forms given by Theorem 1. By choosing $\boldsymbol{L}_1 = \boldsymbol{0}$, the condition in eq. (3) is guaranteed. Then, we can specify a critical point with any $\boldsymbol{V}$ that satisfies the block pattern specified in Theorem 1, i.e., we can choose any $p_i, i = 1,\ldots,r,\bar{p}$ such that $p_i \le m_i$ for $i = 1,\ldots,r$, $\bar{p} \le \bar{m}$ and $\sum_{i=1}^{r} p_i + \bar{p} = \mathrm{rank}(\boldsymbol{A}_2)$. Suppose that $(\boldsymbol{A}_1,\boldsymbol{A}_2)$ is a global minimizer, Proposition 1 gives that $\mathcal{L}(\boldsymbol{A}_1,\boldsymbol{A}_2) = \tfrac{1}{2}\mathrm{Tr}(\boldsymbol{Y}\boldsymbol{Y}^\top) - \tfrac{1}{2}\sum_{i=1}^{r} p_i\sigma_i$. Under the condition that $\min\{d_2,d_1\} \le \sum_{i=1}^{r} m_i$, the global minimum value is achieved by a full rank $\boldsymbol{A}_2$ with $\mathrm{rank}(\boldsymbol{A}_2) = \min\{d_2,d_1\}$ and $p_1 = m_1,\ldots,p_{k-1} = m_{k-1}, p_k = \mathrm{rank}(\boldsymbol{A}_2) - \sum_{i=1}^{k-1} m_i \le m_k$ for some $k \le r$. That is, the singular values are selected in a decreasing order to minimize the function value.

(2): If $(\boldsymbol{A}_2,\boldsymbol{A}_1)$ is a global minimizer and $\min\{d_y,d\} > \sum_{i=1}^{r} m_i$, the global minimum can be achieved by choosing $p_i = m_i$ for all $i = 1,\ldots,r$ and $\bar{p} \ge 0$. In particular, we do not need a full rank $\boldsymbol{A}_2$ to achieve the global minimum. For example, we can choose $\mathrm{rank}(\boldsymbol{A}_2) = \sum_{i=1}^{r} m_i < \min\{d_y,d\}$ with $p_i = m_i$ for all $i = 1,\ldots,r$ and $\bar{p} = 0$.

## PROOF OF PROPOSITION 3

We first prove item 1. Consider a non-optimal-order critical point $(\boldsymbol{A}_1,\boldsymbol{A}_2)$. By Theorem 1, we can write $\boldsymbol{A}_2 = \boldsymbol{U}\boldsymbol{V}\boldsymbol{C}$ where $\boldsymbol{V} = [\mathrm{diag}(\boldsymbol{V}_1,\ldots,\boldsymbol{V}_r,\overline{\boldsymbol{V}}),\boldsymbol{0}]$ and $\boldsymbol{V}_i, i = 1,\ldots,r,\overline{\boldsymbol{V}}$ consist of orthonormal columns. Define the orthonormal block diagonal matrix

$$\boldsymbol{S}^\top := \mathrm{diag}\left(\begin{bmatrix}\boldsymbol{V}_1^\top \\ \boldsymbol{O}_1^\top\end{bmatrix},\cdots,\begin{bmatrix}\boldsymbol{V}_r^\top \\ \boldsymbol{O}_r^\top\end{bmatrix},\begin{bmatrix}\overline{\boldsymbol{V}}^\top \\ \overline{\boldsymbol{O}}^\top\end{bmatrix}\right), \tag{29}$$

where the matrices $\boldsymbol{O}_1,\cdots,\boldsymbol{O}_r,\overline{\boldsymbol{O}}$ are such that each diagonal block forms an orthonormal submatrix. By construction we have $\boldsymbol{S}^\top\boldsymbol{V} = [\mathrm{diag}(\boldsymbol{I}^{m_1\times p_1},\ldots,\boldsymbol{I}^{m_r\times p_r},\boldsymbol{I}^{\bar{m}\times\bar{p}}),\boldsymbol{0}]$, where $\boldsymbol{I}^{m_k\times p_k}$ corresponds to the first $p_k$ columns of the identity matrix $\boldsymbol{I}^{m_k\times m_k}$. Then, $\boldsymbol{A}_2$ can be alternatively written as $\boldsymbol{A}_2 = \boldsymbol{U}\boldsymbol{S}\boldsymbol{S}^\top\boldsymbol{V}\boldsymbol{C}$. Also, denote the columns of $\boldsymbol{U}\boldsymbol{S}$ as

$$\boldsymbol{U}\boldsymbol{S} = [\mathbf{u}_{11}^s,\ldots,\mathbf{u}_{1p_1}^s,\ldots,\mathbf{u}_{r1}^s,\ldots,\mathbf{u}_{rp_r}^s,\bar{\mathbf{u}}_1^s,\ldots,\bar{\mathbf{u}}_{\bar{p}}^s].$$

Since $(\boldsymbol{A}_1,\boldsymbol{A}_2)$ is a non-optimal-order critical point, there exists $1 \le i < j \le r$ such that $p_i < m_i$ and $p_j > 0$. Then, consider the following perturbation of $\boldsymbol{U}\boldsymbol{S}$ for some $\epsilon > 0$.

$$\widetilde{\boldsymbol{M}} = \left[\mathbf{u}_{11}^s,\ldots,\mathbf{u}_{1p_1}^s,\ldots,\frac{\mathbf{u}_{j1}^s+\epsilon\mathbf{u}_{i(p_i+1)}^s}{\sqrt{1+\epsilon^2}},\ldots\bar{\mathbf{u}}_1^s,\ldots,\bar{\mathbf{u}}_{\bar{p}}^s\right],$$

with which we further define the perturbation matrix $\widetilde{A}_2 = \widetilde{M} S^\top V C$. Also, let the perturbation matrix $\widetilde{A}_1$ be generated by eq. (1) with $U \leftarrow \widetilde{M}$ and $V \leftarrow S^\top V$. Note that with this construction, $(\widetilde{A}_1, \widetilde{A}_2)$ satisfies eq. (25), which further implies eq. (27) for $(\widetilde{A}_1, \widetilde{A}_2)$, i.e., $\widetilde{A}_2 \widetilde{A}_1 X = \mathcal{P}_{\mathrm{col}(\widetilde{A}_2)} Y X^\dagger X$. Thus, eq. (28) holds for the point $(\widetilde{A}_1, \widetilde{A}_2)$, and we obtain that

$$
\begin{aligned}
\mathcal{L}(\widetilde{A}_2, \widetilde{A}_1) &= \tfrac{1}{2}\mathrm{Tr}(Y Y^\top) - \tfrac{1}{2}\mathrm{Tr}(\mathcal{P}_{\mathrm{col}(U^\top \widetilde{A}_2)} \Lambda) \\
&= \tfrac{1}{2}\mathrm{Tr}(Y Y^\top) - \tfrac{1}{2}\mathrm{Tr}(\mathcal{P}_{\mathrm{col}(S^\top U^\top \widetilde{A}_2)} S^\top \Lambda S) \\
&= \tfrac{1}{2}\mathrm{Tr}(Y Y^\top) - \tfrac{1}{2}\mathrm{Tr}(\mathcal{P}_{\mathrm{col}(S^\top U^\top \widetilde{M} S^\top V)} S^\top \Lambda S) \\
&= \tfrac{1}{2}\mathrm{Tr}(Y Y^\top) - \tfrac{1}{2}\mathrm{Tr}(\mathcal{P}_{\mathrm{col}(S^\top U^\top \widetilde{M} S^\top V)} \Lambda),
\end{aligned}
$$

where the last equality uses the fact that $S^\top \Lambda S = \Lambda$, as can be observed from the block pattern of $S$ and the multiplicity pattern of $\Lambda$. Also, by the construction of $\widetilde{M}$ and the form of $S^\top V$, a careful calculation shows that only the $i, j$-th diagonal elements of $\mathcal{P}_{\mathrm{col}(S^\top U^\top \widetilde{M} S^\top V)}$ have changed, i.e.,

$$
\left[ \mathcal{P}_{\mathrm{col}(S^\top U^\top \widetilde{M} S^\top V)} \right]_k = \begin{cases} \frac{\epsilon^2}{1+\epsilon^2}, & \text{if } k = i \\ \frac{1}{1+\epsilon^2}, & \text{if } k = j \end{cases}
$$

As the index $i, j$ correspond to the singular values $\sigma_i, \sigma_j$, respectively, and $\sigma_i > \sigma_j$, one obtain that

$$
\mathcal{L}(\widetilde{A}_2, \widetilde{A}_1) = \mathcal{L}(A_2, A_1) - \tfrac{\epsilon^2}{1+\epsilon^2}(\sigma_i - \sigma_j) < \mathcal{L}(A_2, A_1).
$$

Thus, the construction of the point $(\widetilde{A}_2, \widetilde{A}_1)$ achieves a lower function value for any $\epsilon > 0$. Letting $\epsilon \to 0$ and noticing that $\widetilde{M}$ is a perturbation of $U S$, the point $(\widetilde{A}_2, \widetilde{A}_1)$ can be in an arbitrary neighborhood of $(A_2, A_1)$. Lastly, note that $\mathrm{rank}(\widetilde{A}_2) = \mathrm{rank}(A_2)$. This completes the proof of item 1.

Next, we prove item 2. Consider an optimal-order critical point $(A_1, A_2)$. Then, $A_2$ must be non-full rank, since otherwise a full rank $A_2$ with optimal order corresponds to a global minimizer by Proposition 2. Since there exists some $k \leq r$ such that $p_1 = m_1, \ldots, p_{k-1} = m_{k-1}, p_k = \mathrm{rank}(A_2) - \sum_{i=1}^{k-1} m_i \leq m_k$, the necessary form of $A_2$ gives that $A_2 = U V C$ with $V = [\mathrm{diag}(V_1, \ldots, V_k), 0] := [V_{\mathrm{diag}}, 0]$. Using this expression, eq. (1) yields that

$$
A_1 = C^{-1} \left( \begin{bmatrix} (U V_{\mathrm{diag}})^\top Y X^\dagger \\ 0 \end{bmatrix} + C L_1 - \begin{bmatrix} (C L_1)_{1:\mathrm{rank}(A_2),:} X X^\dagger \\ 0 \end{bmatrix} \right).
$$

We now specify our perturbation scheme. Recalling the orthonormal matrix $S$ defined in eq. (29). Then, we consider the following matrices for some $\epsilon_1, \epsilon_2 > 0$

$$
\widetilde{A}_2 = [U V_{\mathrm{diag}}, \epsilon_2 U S_{:,q}, 0] C, \quad \text{where } q = \sum_{i=1}^{k-1} m_i + p_k + 1,
$$

$$
\widetilde{A}_1 = C^{-1} \left( \begin{bmatrix} (U V_{\mathrm{diag}})^\top Y X^\dagger \\ 0 \end{bmatrix} + C L_1 - \begin{bmatrix} (C L_1)_{1:\mathrm{rank}(A_2),:} X X^\dagger \\ \epsilon_1 (U S_{:,q})^\top Y X^\dagger \\ 0 \end{bmatrix} \right).
$$

Our goal is to show that $\mathcal{L}(\widetilde{A}_1, \widetilde{A}_2) < \mathcal{L}(A_1, A_2)$ for $\epsilon_1, \epsilon_2 \to 0$. For this purpose, we need to utilize the condition of critical points in eq. (3), which can be equivalently expressed as

$$
\begin{aligned}
& (I - U V (U V)^\top) Y X^\top L_1^\top C^\top (I - V^\top V) = 0 \\
\overset{(i)}{\Leftrightarrow} \ & \begin{bmatrix} 0 \\ (C L_1)_{(\mathrm{rank}(A_2)+1):d_1,:} X \end{bmatrix} Y^\top (I - U V (U V)^\top) = 0 \\
\Leftrightarrow \ & (C L_1)_{(\mathrm{rank}(A_2)+1):d_1,:} X Y^\top (I - U V (U V)^\top) = 0 \qquad\qquad (30) \\
\overset{(ii)}{\Leftrightarrow} \ & (C L_1)_{(\mathrm{rank}(A_2)+1):d_1,:} X Y^\top (I - U S_{:,1:(q-1)} (U S_{:,1:(q-1)})^\top) = 0 \qquad (31)
\end{aligned}
$$

where (i) follows by taking the transpose and then simplifying, and (ii) uses the fact that $\boldsymbol{V} = \boldsymbol{SS}^\top \boldsymbol{V} = \boldsymbol{S}_{:,1:(q-1)}$ in the case of optimal-order critical point. Calculating the function value at $(\widetilde{\boldsymbol{A}}_1, \widetilde{\boldsymbol{A}}_2)$, we obtain that

$$\mathcal{L}(\widetilde{\boldsymbol{A}}_1, \widetilde{\boldsymbol{A}}_2) = \tfrac{1}{2}\|\underbrace{\boldsymbol{U}\boldsymbol{V}_{\text{diag}}(\boldsymbol{U}\boldsymbol{V}_{\text{diag}})^\top \boldsymbol{Y}\boldsymbol{X}^\dagger \boldsymbol{X}}_{Q}$$

$$+ \underbrace{\epsilon_2 \boldsymbol{U}\boldsymbol{S}_{:,q}(\boldsymbol{C}\boldsymbol{L}_1)_{(\text{rank}(\boldsymbol{A}_2)+1),:}\boldsymbol{X} + \epsilon_1\epsilon_2 \boldsymbol{U}\boldsymbol{S}_{:,q}(\boldsymbol{U}\boldsymbol{S}_{:,q})^\top \boldsymbol{Y}\boldsymbol{X}^\dagger \boldsymbol{X}}_{P} - \boldsymbol{Y}\|_F^2$$

$$= \mathcal{L}(\boldsymbol{A}_1, \boldsymbol{A}_2) + \tfrac{1}{2}[\text{Tr}(\boldsymbol{P}\boldsymbol{P}^\top) + 2\text{Tr}(\boldsymbol{P}\boldsymbol{Q}^\top) - 2\text{Tr}(\boldsymbol{P}\boldsymbol{Y}^\top)].$$

We next simplify the above three trace terms using eq. (31). For the first trace term, observe that

$$\text{Tr}(\boldsymbol{P}\boldsymbol{P}^\top) = \text{Tr}(\epsilon_2^2 \boldsymbol{U}\boldsymbol{S}_{:,q}(\boldsymbol{C}\boldsymbol{L}_1)_{(\text{rank}(\boldsymbol{A}_2)+1),:}\boldsymbol{X}\boldsymbol{X}^\top(\boldsymbol{C}\boldsymbol{L}_1)_{(\text{rank}(\boldsymbol{A}_2)+1),:}^\top(\boldsymbol{U}\boldsymbol{S}_{:,q})^\top)$$

$$+ 2\text{Tr}(\epsilon_1\epsilon_2^2 \boldsymbol{U}\boldsymbol{S}_{:,q}(\boldsymbol{C}\boldsymbol{L}_1)_{(\text{rank}(\boldsymbol{A}_2)+1),:}\boldsymbol{X}\boldsymbol{Y}^\top \boldsymbol{U}\boldsymbol{S}_{:,q}(\boldsymbol{U}\boldsymbol{S}_{:,q})^\top)$$

$$+ \text{Tr}(\epsilon_1^2\epsilon_2^2 \boldsymbol{U}\boldsymbol{S}_{:,q}(\boldsymbol{U}\boldsymbol{S}_{:,q})^\top \boldsymbol{\Sigma}\boldsymbol{U}\boldsymbol{S}_{:,q}(\boldsymbol{U}\boldsymbol{S}_{:,q})^\top)$$

$$\overset{(i)}{=} \text{Tr}(\epsilon_2^2 \boldsymbol{U}\boldsymbol{S}_{:,q}(\boldsymbol{C}\boldsymbol{L}_1)_{(\text{rank}(\boldsymbol{A}_2)+1),:}\boldsymbol{X}\boldsymbol{X}^\top(\boldsymbol{C}\boldsymbol{L}_1)_{(\text{rank}(\boldsymbol{A}_2)+1),:}^\top(\boldsymbol{U}\boldsymbol{S}_{:,q})^\top)$$

$$+ \text{Tr}(\epsilon_1^2\epsilon_2^2 \boldsymbol{U}\boldsymbol{S}_{:,q}(\boldsymbol{U}\boldsymbol{S}_{:,q})^\top \boldsymbol{\Sigma}\boldsymbol{U}\boldsymbol{S}_{:,q}(\boldsymbol{U}\boldsymbol{S}_{:,q})^\top)$$

$$= \epsilon_2^2 \text{Tr}((\boldsymbol{C}\boldsymbol{L}_1)_{(\text{rank}(\boldsymbol{A}_2)+1),:}\boldsymbol{X}\boldsymbol{X}^\top(\boldsymbol{C}\boldsymbol{L}_1)_{(\text{rank}(\boldsymbol{A}_2)+1),:}^\top) + \epsilon_1^2\epsilon_2^2 \text{Tr}(\boldsymbol{S}_{:,q}^\top \boldsymbol{\Lambda}\boldsymbol{S}_{:,q})$$

where (i) follows from eq. (31) as $\boldsymbol{S}_{:,q}$ is orthogonal to the columns of $\boldsymbol{S}_{:,1:(q-1)}$. For the second trace term, we obtain that

$$2\text{Tr}(\boldsymbol{P}\boldsymbol{Q}^\top) = 2\text{Tr}(\epsilon_2 \boldsymbol{U}\boldsymbol{S}_{:,q}(\boldsymbol{C}\boldsymbol{L}_1)_{(\text{rank}(\boldsymbol{A}_2)+1),:}\boldsymbol{X}\boldsymbol{Y}^\top \boldsymbol{U}\boldsymbol{V}_{\text{diag}}(\boldsymbol{U}\boldsymbol{V}_{\text{diag}})^\top)$$

$$+ 2\text{Tr}(\epsilon_1\epsilon_2 \boldsymbol{U}\boldsymbol{S}_{:,q}(\boldsymbol{U}\boldsymbol{S}_{:,q})^\top \boldsymbol{\Sigma}\boldsymbol{U}\boldsymbol{V}_{\text{diag}}(\boldsymbol{U}\boldsymbol{V}_{\text{diag}})^\top)$$

$$= 2\text{Tr}(\epsilon_2 \boldsymbol{U}\boldsymbol{S}_{:,q}(\boldsymbol{C}\boldsymbol{L}_1)_{(\text{rank}(\boldsymbol{A}_2)+1),:}\boldsymbol{X}\boldsymbol{Y}^\top \boldsymbol{U}\boldsymbol{V}_{\text{diag}}(\boldsymbol{U}\boldsymbol{V}_{\text{diag}})^\top)$$

$$+ 2\text{Tr}(\epsilon_1\epsilon_2 \boldsymbol{U}\boldsymbol{S}_{:,q}\boldsymbol{S}_{:,q}^\top \boldsymbol{\Lambda}\boldsymbol{S}\boldsymbol{S}^\top \boldsymbol{V}_{\text{diag}}(\boldsymbol{U}\boldsymbol{V}_{\text{diag}})^\top)$$

$$\overset{(i)}{=} 2\text{Tr}(\epsilon_2 \boldsymbol{U}\boldsymbol{S}_{:,q}(\boldsymbol{C}\boldsymbol{L}_1)_{(\text{rank}(\boldsymbol{A}_2)+1),:}\boldsymbol{X}\boldsymbol{Y}^\top \boldsymbol{U}\boldsymbol{V}_{\text{diag}}(\boldsymbol{U}\boldsymbol{V}_{\text{diag}})^\top)$$

$$+ 2\text{Tr}(\epsilon_1\epsilon_2\sigma_k \boldsymbol{U}\boldsymbol{S}_{:,q}\mathbf{e}_q^\top \boldsymbol{S}^\top \boldsymbol{V}_{\text{diag}}(\boldsymbol{U}\boldsymbol{V}_{\text{diag}})^\top)$$

$$\overset{(ii)}{=} 2\text{Tr}(\epsilon_2 \boldsymbol{U}\boldsymbol{S}_{:,q}(\boldsymbol{C}\boldsymbol{L}_1)_{(\text{rank}(\boldsymbol{A}_2)+1),:}\boldsymbol{X}\boldsymbol{Y}^\top \boldsymbol{U}\boldsymbol{V}_{\text{diag}}(\boldsymbol{U}\boldsymbol{V}_{\text{diag}})^\top),$$

where (i) follows from $\boldsymbol{S}_{:,q}^\top \boldsymbol{\Lambda}\boldsymbol{S} = \sigma_k \mathbf{e}_q^\top$, and (ii) follows from $\mathbf{e}_q^\top \boldsymbol{S}^\top \boldsymbol{V}_{\text{diag}} = \boldsymbol{0}$. For the third trace term, we obtain that

$$2\text{Tr}(\boldsymbol{P}\boldsymbol{Y}^\top) = 2\text{Tr}(\epsilon_2 \boldsymbol{U}\boldsymbol{S}_{:,q}(\boldsymbol{C}\boldsymbol{L}_1)_{(\text{rank}(\boldsymbol{A}_2)+1),:}\boldsymbol{X}\boldsymbol{Y}^\top) + 2\text{Tr}(\epsilon_1\epsilon_2 \boldsymbol{U}\boldsymbol{S}_{:,q}(\boldsymbol{U}\boldsymbol{S}_{:,q})^\top \boldsymbol{\Sigma})$$

$$= 2\text{Tr}(\epsilon_2 \boldsymbol{U}\boldsymbol{S}_{:,q}(\boldsymbol{C}\boldsymbol{L}_1)_{(\text{rank}(\boldsymbol{A}_2)+1),:}\boldsymbol{X}\boldsymbol{Y}^\top) + 2\text{Tr}(\epsilon_1\epsilon_2 \boldsymbol{S}_{:,q}^\top \boldsymbol{\Lambda}\boldsymbol{S}_{:,q}).$$

Combining the expressions for the three trace terms above, we conclude that

$$\tfrac{1}{2}[\text{Tr}(\boldsymbol{P}\boldsymbol{P}^\top) + 2\text{Tr}(\boldsymbol{P}\boldsymbol{Q}^\top) - 2\text{Tr}(\boldsymbol{P}\boldsymbol{Y}^\top)]$$

$$= \tfrac{1}{2}\epsilon_2^2 \text{Tr}((\boldsymbol{C}\boldsymbol{L}_1)_{(\text{rank}(\boldsymbol{A}_2)+1),:}\boldsymbol{X}\boldsymbol{X}^\top(\boldsymbol{C}\boldsymbol{L}_1)_{(\text{rank}(\boldsymbol{A}_2)+1),:}^\top)$$

$$+ (\tfrac{1}{2}\epsilon_1^2\epsilon_2^2 - \epsilon_1\epsilon_2)\text{Tr}(\boldsymbol{S}_{:,q}^\top \boldsymbol{\Lambda}\boldsymbol{S}_{:,q})$$

$$+ 2\epsilon_2 \text{Tr}(\boldsymbol{U}\boldsymbol{S}_{:,q}(\boldsymbol{C}\boldsymbol{L}_1)_{(\text{rank}(\boldsymbol{A}_2)+1),:}\boldsymbol{X}\boldsymbol{Y}^\top[\boldsymbol{U}\boldsymbol{V}_{\text{diag}}(\boldsymbol{U}\boldsymbol{V}_{\text{diag}})^\top - \boldsymbol{I}])$$

$$\overset{(i)}{=} \tfrac{1}{2}\epsilon_2^2 \text{Tr}((\boldsymbol{C}\boldsymbol{L}_1)_{(\text{rank}(\boldsymbol{A}_2)+1),:}\boldsymbol{X}\boldsymbol{X}^\top(\boldsymbol{C}\boldsymbol{L}_1)_{(\text{rank}(\boldsymbol{A}_2)+1),:}^\top)$$

$$+ (\tfrac{1}{2}\epsilon_1^2\epsilon_2^2 - \epsilon_1\epsilon_2)\text{Tr}(\boldsymbol{S}_{:,q}^\top \boldsymbol{\Lambda}\boldsymbol{S}_{:,q})$$

$$= \tfrac{1}{2}\epsilon_2^2 \text{Tr}((\boldsymbol{C}\boldsymbol{L}_1)_{(\text{rank}(\boldsymbol{A}_2)+1),:}\boldsymbol{X}\boldsymbol{X}^\top(\boldsymbol{C}\boldsymbol{L}_1)_{(\text{rank}(\boldsymbol{A}_2)+1),:}^\top) + (\tfrac{1}{2}\epsilon_1^2\epsilon_2^2 - \epsilon_1\epsilon_2)\sigma_k,$$

where (i) follows from eq. (30). Note that the first term in the last equation is nonnegative. Now, letting $\epsilon_2 = \epsilon_1^2 \to 0$, the overall perturbation of the function value becomes

$$\tfrac{1}{2}[\text{Tr}(\boldsymbol{P}\boldsymbol{P}^\top) + 2\text{Tr}(\boldsymbol{P}\boldsymbol{Q}^\top) - 2\text{Tr}(\boldsymbol{P}\boldsymbol{Y}^\top)] = \mathcal{O}(\epsilon_1^4) - \mathcal{O}(\epsilon_1^3) < 0.$$

Thus, the constructed perturbation $(\widetilde{\boldsymbol{A}}_1, \widetilde{\boldsymbol{A}}_2)$ achieves a lower function value, and it can be in an arbitrary neighborhood of $(\boldsymbol{A}_1, \boldsymbol{A}_2)$ as $\epsilon_2 = \epsilon_1^2 \to 0$. Lastly, note that $\mathrm{rank}(\widetilde{\boldsymbol{A}}_2) = \mathrm{rank}(\boldsymbol{A}_2) + 1$.

Next, we prove item 3. We first introduce a technical lemma. Consider row vectors $\mathbf{a}^\top, \mathbf{y}^\top$ with $\mathbf{a}^\top \neq \mathbf{0}$. Define the scalar function $f(\alpha) = \frac{1}{2} \left\| \alpha \mathbf{a}^\top - \mathbf{y}^\top \right\|_2^2$. Then, $f(\alpha)$ is strongly convex as $f''(\alpha) = \|\mathbf{a}\|_2^2 > 0$. Thus, the following fact holds due to strong convexity.

**Fact 1.** *For any $\alpha \in \mathbb{R}$, we can identify a perturbation $\widetilde{\alpha}$ such that $f(\widetilde{\alpha}) > f(\alpha)$.*

Now consider any point $(\boldsymbol{A}_1, \boldsymbol{A}_2) \in \mathcal{X}$. Since $\boldsymbol{A}_2 \boldsymbol{A}_1 \boldsymbol{X} \neq \mathbf{0}$, then there exists a certain row, say, the $i$-th row $(\boldsymbol{A}_2)_{i,:} \boldsymbol{A}_1 \boldsymbol{X}$, that is nonzero. We then apply the above fact with $\alpha = 1, \mathbf{a}^\top = (\boldsymbol{A}_2)_{i,:} \boldsymbol{A}_1 \boldsymbol{X}, \mathbf{y}^\top = \boldsymbol{Y}_{i,:}$, and conclude that one can find a perturbation $\widetilde{\alpha}$ that achieves a higher function value. Equivalently, one can treat the perturbation of $\alpha$ as the perturbation of $(\boldsymbol{A}_2)_{i,:}$, i.e., define the perturbation $(\widetilde{\boldsymbol{A}}_2)_{i,:} := \widetilde{\alpha} (\boldsymbol{A}_2)_{i,:}$.

## PROOF OF THEOREM 3

We first derive the forms of the parameter matrices. Consider a critical point $(\boldsymbol{A}_1, \dots, \boldsymbol{A}_\ell)$ of $\mathcal{L}_D$. By definition of the critical point, we have $\nabla_{\boldsymbol{A}_k} \mathcal{L}_D = 0$ for all $k = 1, \dots, \ell$, which implies that

$$\boldsymbol{A}_{(\ell, k+1)}^\top \boldsymbol{A}_{(\ell, k+1)} \boldsymbol{A}_k \boldsymbol{A}_{(k-1,1)} \boldsymbol{X} (\boldsymbol{A}_{(k-1,1)} \boldsymbol{X})^\top = \boldsymbol{A}_{(\ell, k+1)}^\top \boldsymbol{Y} \boldsymbol{X}^\top \boldsymbol{A}_{(k-1,1)}^\top. \tag{32}$$

Solving this linear system of $\boldsymbol{A}_k$, we obtain that, for some $\boldsymbol{L}_k \in \mathbb{R}^{d_k \times d_{k-1}}$

$$\boldsymbol{A}_k = \boldsymbol{A}_{(\ell, k+1)}^\dagger \boldsymbol{Y} (\boldsymbol{A}_{(k-1,1)} \boldsymbol{X})^\dagger + \boldsymbol{L}_k - \boldsymbol{A}_{(\ell, k+1)}^\dagger \boldsymbol{A}_{(\ell, k+1)} \boldsymbol{L}_k \boldsymbol{A}_{(k-1,1)} \boldsymbol{X} (\boldsymbol{A}_{(k-1,1)} \boldsymbol{X})^\dagger \tag{33}$$

Multiplying eq. (33) on both sides by $\boldsymbol{A}_{(\ell, k+1)}$ on the left and $\boldsymbol{A}_{(k-1,1)} \boldsymbol{X}$ on the right and then simplifying, we obtain that for all $k = 1, \dots, \ell - 1$

$$\boldsymbol{A}_{(\ell,1)} \boldsymbol{X} = \mathcal{P}_{\mathrm{col}(\boldsymbol{A}_{(\ell, k+1)})} \boldsymbol{Y} (\boldsymbol{A}_{(k-1,1)} \boldsymbol{X})^\dagger \boldsymbol{A}_{(k-1,1)} \boldsymbol{X}. \tag{34}$$

On the other hand, applying eq. (32) with $k = \ell$ and multiplying both sides by $\boldsymbol{A}_\ell^\top$ on the right, one obtains that

$$\boldsymbol{A}_{(\ell,1)} \boldsymbol{X} \boldsymbol{X}^\top \boldsymbol{A}_{(\ell,1)}^\top = \boldsymbol{Y} \boldsymbol{X}^\top \boldsymbol{A}_{(\ell,1)}^\top, \tag{35}$$

which, together with eq. (34), further implies that for all $k = 1, \dots, \ell - 1$

$$\mathcal{P}_{\mathrm{col}(\boldsymbol{A}_{(\ell, k+1)})} \boldsymbol{\Sigma}_{k-1} \mathcal{P}_{\mathrm{col}(\boldsymbol{A}_{(\ell, k+1)})} = \boldsymbol{\Sigma}_{k-1} \mathcal{P}_{\mathrm{col}(\boldsymbol{A}_{(\ell, k+1)})}. \tag{36}$$

Following the same argument as that in the proof of Theorem 1, we conclude that $\boldsymbol{A}_{(\ell, k+1)} = \boldsymbol{U}_{k-1} \boldsymbol{V}_{k-1} \boldsymbol{C}_{k-1}$, where $\boldsymbol{U}_{k-1}, \boldsymbol{V}_{k-1}, \boldsymbol{C}_{k-1}$ satisfy the conditions that are stated in the theorem. Then, plugging the expression of $\boldsymbol{A}_{(\ell, k+1)}$ in eq. (33), one obtains the form of $\boldsymbol{A}_k$ in eq. (4).

We next prove the conditions in eq. (6). Note that the first condition is simply a consistency condition on the matrix products. This is because eq. (36) only provides the forms of the matrices $\boldsymbol{A}_{(\ell, k+1)}, k = 1, \dots, l - 1$, which must factorize into the product of individual matrices. For the other condition in eq. (6), note that the proof of eq. (35) uses the weaker condition $(\nabla_{\boldsymbol{A}_\ell} \mathcal{L}_D) \boldsymbol{A}_\ell^\top = \mathbf{0}$ than the original condition $\nabla_{\boldsymbol{A}_\ell} \mathcal{L}_D = \mathbf{0}$ of the critical point. Thus, the forms of the parameter matrices must also satisfy $\nabla_{\boldsymbol{A}_\ell} \mathcal{L}_D = \mathbf{0}$, i.e., $\boldsymbol{A}_{(\ell,1)} \boldsymbol{X} (\boldsymbol{A}_{(\ell-1,1)} \boldsymbol{X})^\top = \boldsymbol{Y} \boldsymbol{X}^\top \boldsymbol{A}_{(\ell-1,1)}^\top$. Then, plugging eq. (34) in the above condition and simplifying, one obtains eq. (6).

## PROOF OF PROPOSITION 4

Note that by expansion $\mathcal{L}_D = \frac{1}{2} \mathrm{Tr}(\boldsymbol{Y} \boldsymbol{Y}^\top) - \mathrm{Tr}(\boldsymbol{A}_{(\ell,1)} \boldsymbol{X} \boldsymbol{Y}^\top) + \frac{1}{2} \mathrm{Tr}(\boldsymbol{A}_{(\ell,1)} \boldsymbol{X} \boldsymbol{X}^\top \boldsymbol{A}_{(\ell,1)}^\top)$. For any $\boldsymbol{A}_1, \dots, \boldsymbol{A}_\ell$ that satisfy eq. (32), we have shown that they must satisfy eq. (34) with $k = 1$, with which we further obtain that

$$\begin{aligned} \mathcal{L} &= \tfrac{1}{2} \mathrm{Tr}(\boldsymbol{Y} \boldsymbol{Y}^\top) - \tfrac{1}{2} \mathrm{Tr}(\mathcal{P}_{\mathrm{col}(\boldsymbol{A}_{(\ell,2)})} \boldsymbol{\Sigma}_0) \\ &= \tfrac{1}{2} \mathrm{Tr}(\boldsymbol{Y} \boldsymbol{Y}^\top) - \tfrac{1}{2} \mathrm{Tr}(\mathcal{P}_{\mathrm{col}(\boldsymbol{U}_0^\top \boldsymbol{A}_{(\ell,2)})} \boldsymbol{\Lambda}_0). \end{aligned} \tag{37}$$

Consider a critical point $(\boldsymbol{A}_1, \ldots, \boldsymbol{A}_\ell)$ so that eq. (37) holds. Using the form of critical points $\boldsymbol{A}_{(\ell,2)} = \boldsymbol{U}_0\boldsymbol{V}_0\boldsymbol{C}_0$, eq. (37) further becomes

$$\begin{aligned}
\mathcal{L} &= \tfrac{1}{2}\mathrm{Tr}(\boldsymbol{Y}\boldsymbol{Y}^\top) - \tfrac{1}{2}\mathrm{Tr}(\,\mathcal{P}_{\mathrm{col}(\boldsymbol{V}_0\boldsymbol{C}_0)}\boldsymbol{\Lambda}_0) \\
&= \tfrac{1}{2}\mathrm{Tr}(\boldsymbol{Y}\boldsymbol{Y}^\top) - \tfrac{1}{2}\mathrm{Tr}(\boldsymbol{V}_0^\top\boldsymbol{\Lambda}_0\boldsymbol{V}_0) \\
&\stackrel{(i)}{=} \tfrac{1}{2}\mathrm{Tr}(\boldsymbol{Y}\boldsymbol{Y}^\top) - \tfrac{1}{2}\sum_{i=1}^{r(0)} p_i(0)\sigma_i(0),
\end{aligned}$$

where (i) utilizes the block pattern of $\boldsymbol{V}_0$ and the multiplicity pattern of $\boldsymbol{\Lambda}_0$ that are specified in Theorem 3.

## PROOF OF PROPOSITION 5

Observe that the product matrix $\boldsymbol{A}_{(\ell,2)}$ is equivalent to the class of matrices $\boldsymbol{B}_2 \in \mathbb{R}^{\min\{d_\ell, \ldots, d_2\} \times d_1}$. Consider a critical point $(\boldsymbol{B}_2, \boldsymbol{A}_1)$ of the shallow linear network $\mathcal{L} := \tfrac{1}{2}\|\boldsymbol{B}_2\boldsymbol{A}_1\boldsymbol{X} - \boldsymbol{Y}\|_F^2$. By Proposition 2, we conclude that

- If $\min\{d_\ell, \ldots, d_1\} \le \sum_{i=1}^{r(0)} m_i(0)$, then $(\boldsymbol{A}_1, \boldsymbol{B}_2)$ is a global minimizer if and only if $\boldsymbol{B}_2$ is full rank and $p_1 = m_1(0), \ldots, p_{k-1} = m_{k-1}(0), p_k = \mathrm{rank}(\boldsymbol{B}_2) - \sum_{i=1}^{k-1} m_i(0) \le m_k(0)$ for some $k \le r(0)$;

- If $\min\{d_\ell, \ldots, d_1\} > \sum_{i=1}^{r(0)} m_i(0)$, then $(\boldsymbol{A}_1, \boldsymbol{B}_2)$ is a global minimizer if and only if $p_i = m_i(0)$ for all $i = 1, \ldots, r(0)$ and $\bar{p}(0) \ge 0$. In particular, $\boldsymbol{B}_2$ can be non-full rank with $\mathrm{rank}(\boldsymbol{B}_2) = \sum_{i=1}^{r(0)} m_i(0)$.

Note that $\mathcal{L}_D$ achieves the same global minimum as $\mathcal{L}$. Hence Proposition 1 and Proposition 4 must match, which yields that $\sum_{i=1}^{r(0)} p_i\sigma_i(0) = \sum_{i=1}^{r(0)} p_i(0)\sigma_i(0)$. We then conclude that $p_i(0) = p_i$ for $i = 1, \ldots, r(0)$ as $\sigma_1(0) > \cdots > \sigma_{r(0)}(0)$. This proves the proposition.

## PROOF OF THEOREM 4

The proof is similar to that for shallow linear networks. Consider a deep-non-optimal-order critical point $(\boldsymbol{A}_1, \ldots, \boldsymbol{A}_\ell)$, and define the orthonormal block matrix $\boldsymbol{S}_k$ using the blocks of $\boldsymbol{V}_k$ in a similar way as eq. (29). Then, $\boldsymbol{A}_{(k,k+2)}$ takes the form $\boldsymbol{A}_{(l,k+2)} = \boldsymbol{U}_k\boldsymbol{S}_k\boldsymbol{S}_k^\top\boldsymbol{V}_k\boldsymbol{C}_k$. Since $\boldsymbol{A}_{(l,k+2)}$ is of non-optimal order, there exists $i < j < r(k)$ such that $p_i(k) < m_i(k)$ and $p_j(k) > 0$. Thus, we perturb the $j$-th column of $\boldsymbol{U}_k\boldsymbol{S}_k$ to be $\frac{\mathbf{u}_{j1}^s + \epsilon\mathbf{u}_{i(p_i(k)+1)}^s}{\sqrt{1+\epsilon^2}}$, and denote the resulting matrix as $\widetilde{\boldsymbol{M}}_k$. Then, we perturb $\boldsymbol{A}_\ell$ to be $\widetilde{\boldsymbol{A}}_\ell = \widetilde{\boldsymbol{M}}_k(\boldsymbol{U}_k\boldsymbol{S}_k)^\top\boldsymbol{A}_\ell$ so that $\widetilde{\boldsymbol{A}}_\ell\boldsymbol{A}_{(\ell-1,k+2)} = \widetilde{\boldsymbol{M}}_k\boldsymbol{S}_k^\top\boldsymbol{V}_k\boldsymbol{C}_k$. Moreover, we generate $\widetilde{\boldsymbol{A}}_{k+1}$ by eq. (4) with $\boldsymbol{U}_k \leftarrow \widetilde{\boldsymbol{M}}_k, \boldsymbol{V}_k \leftarrow \boldsymbol{S}_k^\top\boldsymbol{V}_k$. Note that such construction satisfies eq. (32), and hence also satisfies eq. (34), which further yields that

$$\widetilde{\boldsymbol{A}}_\ell\boldsymbol{A}_{(\ell-1,k+2)}\widetilde{\boldsymbol{A}}_{k+1}\boldsymbol{A}_{(k,1)}\boldsymbol{X} = \mathcal{P}_{\mathrm{col}(\widetilde{\boldsymbol{A}}_\ell\boldsymbol{A}_{(\ell-1,k+2)})}\boldsymbol{Y}(\boldsymbol{A}_{(k,1)}\boldsymbol{X})^\dagger\boldsymbol{A}_{(k,1)}\boldsymbol{X}.$$

With the above equation, the function value at this perturbed point is evaluated as

$$\mathcal{L}(\widetilde{\boldsymbol{A}}_\ell, \ldots, \widetilde{\boldsymbol{A}}_{k+1}, \ldots) = \tfrac{1}{2}\mathrm{Tr}(\boldsymbol{Y}\boldsymbol{Y}^\top) - \tfrac{1}{2}\mathrm{Tr}(\,\mathcal{P}_{\mathrm{col}(\boldsymbol{S}_k^\top\boldsymbol{U}_k^\top\widetilde{\boldsymbol{M}}_k\boldsymbol{S}_k^\top\boldsymbol{V}_k)}\boldsymbol{\Lambda}_k).$$

Then, a careful calculation shows that only the $i, j$-th diagonal elements of $\mathcal{P}_{\mathrm{col}(\boldsymbol{S}_k^\top\boldsymbol{U}_k^\top\widetilde{\boldsymbol{M}}_k\boldsymbol{S}_k^\top\boldsymbol{V}_k)}\boldsymbol{\Lambda}$ have changed, and are $\frac{\epsilon^2}{1+\epsilon^2}, \frac{1}{1+\epsilon^2}$, respectively. We then conclude that

$$\mathcal{L}(\boldsymbol{A}_1, \ldots, \widetilde{\boldsymbol{A}}_{k+1}, \ldots, \widetilde{\boldsymbol{A}}_\ell) = \mathcal{L}(\boldsymbol{A}_1, \ldots, \boldsymbol{A}_\ell) - \tfrac{\epsilon^2}{1+\epsilon^2}(\sigma_i(k) - \sigma_j(k)) < \mathcal{L}(\boldsymbol{A}_1, \ldots, \boldsymbol{A}_\ell).$$

Now consider a deep-optimal-order critical point $(\boldsymbol{A}_1, \ldots, \boldsymbol{A}_\ell)$. Note that with $\boldsymbol{A}_{(\ell-2,1)}$ fixed to be a constant, the deep linear network reduces to a shallow linear network with parameters $(\boldsymbol{A}_\ell, \boldsymbol{A}_{\ell-1})$. Since $(\boldsymbol{A}_\ell, \boldsymbol{A}_{\ell-1})$ is not a non-global minimum critical point of this shallow linear network and $\boldsymbol{A}_\ell$

is of optimal-order, we can apply the perturbation scheme in the proof of Proposition 3 to identify a perturbation $(\widetilde{\boldsymbol{A}}_\ell, \widetilde{\boldsymbol{A}}_{\ell-1})$ with $\text{rank}(\widetilde{\boldsymbol{A}}_\ell) = \text{rank}(\boldsymbol{A}_\ell) + 1$ that achieves a lower function value.

Consider any point in $\mathcal{X}_D$. Since $\boldsymbol{A}_{(\ell,1)}\boldsymbol{X} \neq \boldsymbol{0}$, we can scale the nonzero row, say, the $i$-th row $(\boldsymbol{A}_\ell)_{i,:}\boldsymbol{A}_{(\ell-1,1)}\boldsymbol{X}$ properly in the same way as that in the proof of Proposition 3 to increase the function value. Lastly, item 1 and item 2 imply that every local minimum is a global minimum for these two types of critical points. Moreover, combining items 1,2 and 3, we conclude that every critical point of these two types in $\mathcal{X}_D$ is a saddle point.

