# OpenReview forum: "Critical Points of Linear Neural Networks: Analytical Forms and Landscape Properties"
_ICLR.cc/2018/Conference — Accept (Poster)_

### Official Review · AnonReviewer2 · 2017-11-20
**Authors of this paper provided full characterization of the analytical forms of the critical points for the square loss function of three types of neural networks: shallow linear networks, deep linear networks and shallow ReLU nonlinear networks.**

**Rating:** 7
**Confidence:** 3

**Review:**

Authors of this paper provided full characterization of the analytical forms of the critical points for the square loss function of three types of neural networks: shallow linear networks, deep linear networks and shallow ReLU nonlinear networks. The analytical forms of the critical points have direct implications on the values of the corresponding loss functions, achievement of global minimum, and various landscape properties around these critical points.

The paper is well organized and well written. Authors exploited the analytical forms of the critical points to provide a new proof for characterizing the landscape around the critical points. This technique generalizes existing work under full relaxation of assumptions. In the linear network with one hidden layer, it generalizes the work Baldi & Hornik (1989) with arbitrary network parameter dimensions and any data matrices; In the deep linear networks, it generalizes the result in Kawaguchi (2016) under no assumptions on the network parameters and data matrices. Moreover, it also provides new characterization for shallow ReLU nonlinear networks, which is not discussed in previous work.

The results obtained from the analytical forms of the critical points are interesting, but one problem is that how to obtain the proper solution of equation (3)? In the Example 1, authors gave a concrete example to demonstrate both local minimum and local maximum do exist in the shallow ReLU nonlinear networks by properly choosing these matrices satisfying (12). It will be interesting to see how to choose these matrices for all the studied networks with some concrete examples.

---

> ### Author Response · Authors · 2017-12-14
> **We thank the reviewer for providing valuable feedbacks. Below is a point-to-point response.**
>
> Q1: How to obtain the proper solution of eq (3)?
>
> A: We note that matrix L_1 can be chosen arbitrarily. Thus, choosing L_1 = 0 always satisfies eq (3), and we obtain the critical points A1 = C^-1 V^t U^t YX^+, A_2 = UVC with any invertible matrix C and any matrix V with the structure specified in Theorem 1. To further obtain the solution of eq (3) with nonzero L_1, one can fix a proper V and solve the linear equation on C in eq (3). If a solution exists, we then obtain the form of a corresponding critical point.
>
> Q2: In Example 1, authors gave a concrete example to demonstrate both local minimum and local maximum do exist in the shallow ReLU nonlinear networks by property choosing these matrices satisfying (12). How to choose these matrices for all the studied networks with some concrete examples?
>
> A: For other studied networks (shallow linear and deep linear networks), examples can be constructed based on the corresponding characterizations. For shallow linear networks, as our response for Q1, we can set L_1 = 0 (so that eq (3) is satisfied), and then A1 = C^-1 V^t U^t YX^+, A_2 = UVC with any invertible matrix C and any matrix V with the structure specified in Theorem 1 are critical points. Furthermore, if we further set the parameters p_i according to Prop 2, we obtain examples for global minima. For deep linear networks, it is also easier to construct examples by setting L_k = 0 for all k so that eq (6) is satisfied, and we can then obtain critical points for any invertible C_k and proper V_k with the structure specified in Theorem 3.  Furthermore, if we further set the parameters p_i(0) according to Prop 5, we obtain examples for global minima. We note that all local minima are also global minima for these linear networks.

---

### Official Review · AnonReviewer3 · 2017-11-26
**This paper studies the critical points of shallow and deep linear networks. The authors give a (necessary and sufficient) characterization of the form of critical points and use this to derive necessary and sufficient conditions for which critical points are global optima. While the exposition of the paper can be improved in my view this is a neat and concise result and merits publication in ICLR.**

**Rating:** 7
**Confidence:** 5

**Review:**

This paper studies the critical points of shallow and deep linear networks. The authors give a (necessary and sufficient) characterization of the form of critical points and use this to derive necessary and sufficient conditions for which critical points are global optima. Essentially this paper revisits a classic paper by Baldi and Hornik (1989) and relaxes a few requires assumptions on the matrices. I have not checked the proofs in detail but the general strategy seems sound. While the exposition of the paper can be improved in my view this is a neat and concise result and merits publication in ICLR. The authors also study the analytic form of critical points of a single-hidden layer ReLU network. However, given the form of the necessary and sufficient conditions the usefulness of of these results is less clear.


Detailed comments:

- I think in the title/abstract/intro the use of Neural nets is somewhat misleading as neural nets are typically nonlinear. This paper is mostly about linear networks. While a result has been stated for single-hidden ReLU networks. In my view this particular result is an immediate corollary of the result for linear networks. As I explain further below given the combinatorial form of the result, the usefulness of this particular extension to ReLU network is not very clear. I would suggest rewording title/abstract/intro

- Theorem 1 is neat, well done!

- Page 4 p_i’s in proposition 1
From my understanding the p_i have been introduced in Theorem 1 but given their prominent role in this proposition they merit a separate definition (and ideally in terms of the A_i directly).

- Theorems 1, prop 1, prop 2, prop 3, Theorem 3, prop 4 and 5
	Are these characterizations computable i.e. given X and Y can one run an algorithm to find all the critical points or at least the parameters used in the characterization p_i, V_i etc?

- Theorems 1, prop 1, prop 2, prop 3, Theorem 3, prop 4 and 5
	Would recommend a better exposition why these theorems are useful. What insights do you gain by knowing these theorems etc. Are less sufficient conditions that is more intuitive or useful. (an insightful sufficient condition in some cases is much more valuable than an unintuitive necessary and sufficient one).

- Page 5 Theorem 2
	Does this theorem have any computational implications? Does it imply that the global optima can be found efficiently, e.g. are saddles strict with a quantifiable bound?

- Page 7 proposition 6 seems like an immediate consequence of Theorem 1 however given the combinatorial nature of the K_{I,J} it is not clear why this theorem is useful. e.g . back to my earlier comment w.r.t. Linear networks given Y and X can you find the parameters of this characterization with a computationally efficient algorithm?

---

> ### Author Response · Authors · 2017-12-14
> **We thank the reviewer for providing valuable feedbacks. Below is a point-to-point response.**
>
> Q1:  I think in the title/abstract/intro the use of neural nets is somewhat misleading as neural nets are typically nonlinear. This paper is mostly about linear networks.  I would suggest rewording title/abstract/intro.
>
> A: We agree, and we will reword neural networks as linear neural networks.
>
> Q2: From my understanding, the p_i have been introduced in Theorem 1 but given their prominent role in this proposition they merit a separate definition.
>
> A: We will separately define p_i, and further clarify their impact on the forms of the A_i. The p_i's in Theorem 1 can be any positive integer smaller than the corresponding m_i (i.e., the multiplicity of singular value sigma_i), and the sum of the pi's is equal to the rank of A_2.
>
> Q3: Given X and Y, can one run an algorithm to find all the critical points or at least the parameters used in the characterization p_i, V_i etc?
>
> A: We first note that in general, the set of critical points is uncountable and cannot be fully listed out. Hence, the characterization of the analytical forms of the critical points is more important in terms of its analytical structure, which can have a direct implication on the global optimality conditions and can be exploited to prove the landscape properties.
>
> On the other hand, these forms do suggest ways to obtain some critical points. For shallow linear networks, if we choose L_1 = 0 (i.e., eq (3) is satisfied), we directly obtain the form of critical points A1 = C^-1 V^t U^t YX^+, A_2 = UVC, where C is any invertible matrix and V is any matrix with the structures specified in Theorem 1. For nonzero L_1, one can fix a proper V and solve the linear equation on C in eq (3). If a solution exists, we then obtain the form of a corresponding critical point. For shallow ReLU networks, one can find the solution of parameters from eq (12) following the same procedure as the above, and one needs to further verify the existence conditions in eqs (13, 14). For deep linear networks, in the case where L_k = 0 for all k (i.e., eq (6) is satisfied), we can obtain the form of critical points for any invertible C_k and proper V_k. For nonzero L_k, eq (6) needs to be verified for given C_k and V_k to determine a critical point.
>
> Q4: What insights do you gain by knowing Theorems 1, prop 1, prop 2, prop 3, Theorem 3, prop 4 and 5?
>
> A: The analytical forms in Theorems 1,3 help to characterize the global optima in Prop 2,5. They also help to identify the descent and ascent directions at critical points in Prop 3 and Theorem 4, establishing the landscape properties around them. Such properties then provide an alternative approach to show the equivalence between local minima and global minima.
>
> For further insights, Prop. 2 case 1 implies that the parameter matrix A_2 must be full rank at any global minima. In particular, A_1 is also full rank at global minima in this case under the assumptions on data matrices and network dimensions in Baldi & Hornik (1989).  Similar conclusions hold for deep linear networks, e.g., if all the parameter matrices are square and the data matrices satisfy the assumptions in Baldi & Hornik (1989), then all global minima must correspond to full rank parameter matrices.
>
> Q5: Does Theorem 2 have any computational implications, e.g. are saddles strict with a quantifiable bound?
>
> A: Theorem 2 does not directly imply the strictness of saddle points. In fact, the saddle points in Theorem 2 can be non-strict for arbitrary data X and Y (the case we consider). As an example, consider the loss of the linear network L(a_2, a_1) = (y-a_2 a_1 x)^2, where a_1, a_2, x and y are all scalars. Consider the case with y=0, then L(a_2, a_1) = (a_2 a_1 x)^2. One can check that the Hessian at the saddle point a_1 = 0, a_2 = 1 is [2x^2, 0; 0, 0], which does not have negative eigenvalue. Thus, non-strict saddle can exist if data are arbitrary.
>
> Q6: Why is Proposition 6 useful, can you find the parameters of this characterization with a computationally efficient algorithm?
>
> A: Prop 6 is more useful in terms of the structure of the forms it characterizes for the critical points. For example, such forms in Prop 6 (and its special case of Prop 7) are exploited to construct a spurious local minimum in Example 1. Computationally, as pointed out in our response to Q3, we can compute/verify the parameters for various cases, but we cannot fully list all critical points, which are uncountable.

---

> > ### Comment · AnonReviewer3 · 2018-01-12
> > **Reply**
> >
> > I am satisfied with the authors response and maintain my rating and acceptance recommendation.

---

### Official Review · AnonReviewer1 · 2017-11-28
**An interesting work on the characterization of critical points of neural networks**

**Rating:** 6
**Confidence:** 4

**Review:**

This paper mainly focuses on the square loss function of linear networks. It provides the sufficient and necessary characterization for the forms of critical points of one-hidden-layer linear networks. Based on this characterization, the authors are able to discuss different types of non-global-optimal critical points and show that every local minimum is a global minimum for one-hidden-layer linear networks. As an extension, the manuscript also characterizes the analytical forms for the critical points of deep linear networks and deep ReLU networks, although only a subset of non-global-optimal critical points are discussed. In general, this manuscript is well written.

Pros:
1. This manuscript provides the sufficient and necessary characterization of critical points for deep networks.
2. Compared to previous work, the current analysis for one-hidden-layer linear networks doesn’t require assumptions on parameter dimensions and data matrices. The novel analyses, especially the technique to characterize critical points and the proof of item 2 in Proposition 3, will probably be interesting to the community.
3. It provides an example when a local minimum is not global for a one-hidden-layer neural network with ReLU activation.

Cons:
1. I'm concerned that the contribution of this manuscript is a little incremental. The equivalence of global minima and local minima for linear networks is not surprising based on existing works e.g. Hardt & Ma (2017) and Kawaguchi (2016).
2. Unlike one-hidden-layer linear networks, the characterizations of critical points for deep linear networks and deep ReLU networks seem to be hard to be interpreted. This manuscript doesn't show that every local minimum of these two types of deep networks is a global minimum, which actually has been shown by existing works like Kawaguchi (2016) with some assumptions. The behaviors of linear networks and practical (deep and nonlinear) networks are very different. Under such circumstance, the results about one-hidden-layer linear networks are less interesting to the deep learning community.

Minors:
There are some mixed-up notations: tilde{A_i} => A_i , and rank(A_2) => rank(A)_2 in Proposition 3.

---

> ### Author Response · Authors · 2017-12-14
> **We thank the reviewer for providing valuable feedbacks. Below is a point-to-point response.**
>
> Q1: Contribution of this manuscript is a little incremental. Equivalence of global minima and local minima for linear networks is not surprising, e.g. Hardt & Ma (2017) and Kawaguchi (2016).
>
> A: We agree that the equivalence between global minima and local minima for linear networks has been established in the existing works. This work was in fact highly inspired by these previous results. However, the focus of this paper is different. The main results lie in providing the analytical forms of critical points for linear networks and ReLU networks, which further provides analytical forms of global optima (Prop. 2, 5) for linear networks and shows the existence of spurious local minima (Example 1) for ReLU networks. This type of results were not in Hardt & Ma (2017) and Kawaguchi (2016). We then further exploit such analytical forms of critical points to provide alternative arguments for the equivalence between local minima and global minima for linear networks, which were originally established in Kawaguchi (2016) by exploiting the necessary conditions of local minima.
>
> Q2: The characterizations of critical points for deep linear and ReLU networks seem to be hard to be interpreted.
>
> A: We agree that the forms of critical points for deep linear and ReLU networks are complex. But they can still be useful for various cases. For example, the characterization of critical points for deep linear networks in Theorem 3 further helps to characterize the global minima in Prop 5, and the characterization of critical points for ReLU networks in Prop 7 further helps to show the existence of spurious local minima in Example 1.
>
> Q3: This manuscript doesn't show that every local minimum of these two types of deep networks (i.e., deep linear and ReLu networks) is a global minimum, which actually has been shown by Kawaguchi (2016) with some assumptions.
>
> A: Indeed, under some assumptions, Kawaguchi (2016) established the equivalence between local minima and global minima for both deep linear and ReLU networks. However, Kawaguchi (2016) assumed that each ReLU is activated according to Bernoulli distribution, and studied the expected loss over the randomness of the activations. In this setting, the loss function of ReLU networks reduces to that of linear networks. In comparison, we neither assume nor average over the randomness of the activations in our loss for ReLU networks. In fact, we showed that spurious local minima do exist for such loss of ReLU networks (Example 1).
>
> Q4: The behaviors of linear networks and practical deep and nonlinear networks are very different. The results about one-hidden-layer linear networks are less interesting to the deep learning community.
>
> A: We agree that it is challenging to understand deep and nonlinear networks, and their behaviors can be very different from shallow linear networks. Ultimately, we agree that tools for studying shallow linear networks won’t be sufficient. However, understanding shallow linear networks can still be beneficial in various cases. For example, our characterizations of deep linear and shallow ReLU networks are further developments of the characterizations of shallow linear networks. Such understandings allow us to show the existence of spurious local minimum for ReLU networks (Example 1), which is different from the behavior of linear networks.
>
> We also thank the reviewer for pointing out the mixed-up notations. We will fix these notations.

---

> > ### Comment · AnonReviewer1 · 2018-01-12
> > **Reply**
> >
> > Thanks for the clarification. Most of my concerns are addressed. An anonymous reviewer raised a concern about the overlap with existing work, Li et al. 2016b. The authors' comments about this related work sound ok to me. But I would suggest the authors add more discussion about it. Overall the paper is above the acceptance threshold in my opinion and I keep my rating.

---

### Author Response · Authors · 2017-12-31
**Revision**

Based on the reviewers' comments, we uploaded a revision that made the following changes. We are happy to make further changes if the reviewers have additional comments.

1. We fixed the mixed-up notations in Prop. 3. Note that in item 3 of Prop. 3, we only perturb A_2 to tilde{A_2}.

2. In the title, abstract and introduction, we reworded neural networks as linear neural networks whenever applicable.

3. We added Remark 1 above Prop. 1 to separately define the parameters p_i's.

4. In the paragraph before Remark 1, we commented that the critical points characterized in Theorem 1 cannot be fully listed out because they are in general uncountable. We also explained how to use the form in Theorem 1 to obtain some critical points. We also note that the analytical structure of the critical points is important, which determines the landscape properties of the loss function. This comment is also applicable to the case of deep linear networks and shallow ReLU networks.

5. Towards the end of the paragraph after Prop. 2, we added further insight of Prop. 2 in a special case, i.e., both A_2 and A_1 are full rank at global minima under the assumptions on data matrices and network dimensions in Baldi & Hornik (1989). In the paragraph after Prop. 5, we added the similar understanding, i.e., if all the parameter matrices are square and the data matrices satisfy the assumptions as in Baldi & Hornik (1989), then all global minima must correspond to full rank parameter matrices.

6. After Theorem 2, we commented that the saddle points can be non-strict for arbitrary data matrices X and Y with an illustrative example.

7. We added another related work Li et al. 2016b.

---

### Decision · Program_Chairs · 2018-01-29
**ICLR 2018 Conference Acceptance Decision**

**Decision:**

Accept (Poster)

**Comment:**

I recommend acceptance based on the positive reviews. The paper analyzes critical points for linear neural networks and shallow ReLU networks. Getting characterization of critical points for shallow ReLU networks is a great first step.